# BAN&PICK: ENHANCING PERFORMANCE AND EFFICIENCY OF MOE-LLMS VIA SMARTER ROUTING

## ABSTRACT

Sparse Mixture-of-Experts (MoE) has become a key architecture for scaling large language models (LLMs) efficiently. Recent fine-grained MoE designs introduce hundreds of experts per layer, with multiple experts activated per token, enabling stronger specialization. However, during pre-training, routers are optimized mainly for stability and robustness: they converge prematurely and enforce balanced usage, limiting the full potential of model performance and efficiency at inference. In this work, we uncover two overlooked issues: (i) a few highly influential experts are underutilized due to premature and balanced routing decisions; and (ii) enforcing a fixed number of active experts per token introduces substantial redundancy. Instead of retraining models or redesigning MoE architectures, we introduce **Ban&Pick**, a post-training, plug-and-play strategy for smarter routing. **Pick** discovers and reinforces key experts—a small group with outsized impact on performance—leading to notable accuracy gains across domains. **Ban** further dynamically prunes redundant experts based on layer and token sensitivity, delivering faster inference with minimal accuracy loss. Experiments on fine-grained MoE-LLMs (DeepSeek, Qwen3) across math, code, and general reasoning benchmarks demonstrate that Ban&Pick delivers free performance gains and inference acceleration without retraining or architectural changes. For instance, on Qwen3-30B-A3B, it improves accuracy from 80.67 to 84.66 on AIME2024 and from 65.66 to 68.18 on GPQA-Diamond, while accelerating inference by 1.25× under the `vLLM`.

## 1 INTRODUCTION

Large language models (LLMs) have achieved remarkable progress (Zhou et al., 2024), yet the ever-growing demand for higher capacity presents significant challenges for efficiency. As a promising solution, Mixture-of-Experts (MoE) architectures scale parameter count without proportionally increasing computation, by activating a small subset of experts for each input (Artetxe et al., 2022).

Recent advances have pushed MoE towards a fine-grained design (Dai et al., 2024), where each layer contains a large number of relatively small experts. For example, DeepSeek-v2.5 (DeepSeek-AI et al., 2024) employs 160 experts per layer, while Qwen3-MoE (Yang et al., 2025) adopts 128 experts per layer. Unlike coarse-grained MoE (Jiang et al., 2024), which typically consists of a few large experts derived from the same base model, fine-grained MoE initializes each expert independently and randomly. This design promotes specialization: experts gradually develop distinct skills, excelling in areas such as math, code, or general reasoning. Such strong differentiation suggests that fine-grained MoE has the potential to leverage expert diversity more effectively.

However, we argue that a critical but overlooked phenomenon exists: **the routing strategies learned during pre-training cannot fully exploit this specialization**. Specifically, we empirically discover that current routing strategies inadvertently limit the potential to leverage the most impactful experts while simultaneously introducing redundancy where many selected experts contribute minimally to the final output. Instead of attempting to redesign training objectives or expert architectures, we take a post-training perspective, aiming to unlock this untapped potential efficiently and effectively.

Delving into this question, we surprisingly find that within the most frequently selected experts of fine-grained MoE models, a small fraction plays an outsized role: simply forcing these experts to be chosen for every input can noticeably boost accuracy on certain datasets. Motivated by this striking observation, we conduct an empirical analysis of expert usage across domains and discover that

each domain consistently relies on its own set of frequently selected experts, which we refer to as domain-specialized experts. Yet, a closer examination further shows that only a tight subset of these experts has a decisive impact on the final logits distribution, while the influence of the others remains marginal. We term these most critical ones key experts. Our **Pick** module is designed to reinforce their influence during routing, getting performance gains by emphasizing these key experts.

In parallel, we aim to alleviate redundancy in expert activation. Under the constraint that a fixed number of experts must be selected for each token, many activated experts contribute little to the final output, reflecting the redundancy introduced by less relevant experts. Our **Ban** module jointly considers layer sensitivity and token sensitivity to assess the redundancy in expert selection, and applies dynamic pruning of experts during inference. In this way, Ban reduces unnecessary computation and achieves significant inference acceleration with only minor accuracy loss.

Attempting to bridge the gap between routing strategies learned during pre-training—which prioritizes stability and balance—and optimal inference-time expert utilization, Pick and Ban empower truly impactful experts to contribute more while reducing participation from low-contribution experts:

- **Pick** amplifies excellence by identifying experts that demonstrate superior performance on specific domains and strategically increasing their influence during inference, maximizing effectiveness.
- **Ban** eliminates inefficiency by dynamically excluding experts that contribute minimally during inference, reducing computational costs while preserving essential capabilities.

Together, we introduce Ban&Pick, a unified framework that optimizes expert utilization in inference to improve both the accuracy and efficiency of fine-grained MoE without retraining or architectural changes. Evaluated on five widely used datasets spanning math, code, and general reasoning, the results demonstrate that on the Qwen3 series models, applying Pick alone brings an average 2.83% performance improvement. By further incorporating Ban, the joint Ban&Pick strategy achieves an average 1.41% improvement while simultaneously accelerating inference by 1.25× under the `vLLM`.

## 2 RELATED WORKS

**Enhancing Expert Utilization in MoE.** Most existing work attempts to improve expert utilization efficiency in MoE models by redesigning training objectives or expert architectures during pre-training (Yan et al., 2025; Huang et al., 2024a; Jin et al., 2024). However, while these approaches are insightful, they often sacrifice training stability for flexibility. More recently, (Wang et al., 2025a) proposes a post-training method specifically designed for MoE models that employ the `<think>` token, enhancing reasoning by amplifying the weights of Cognitive Experts.

**Dynamic Expert Pruning of MoE.** Dynamic expert pruning has emerged as a key technique for accelerating MoE inference. Lu et al. (2024) proposed a pruning strategy that simply skips experts with less weights. Building on this, Huang et al. (2024b) introduced a critical token protection mechanism. However, the substantial sensitivity differences across layers of MoE models and across tokens in long chain-of-thought reasoning are often overlooked, leaving room for further optimization.

## 3 EMPIRICAL FINDINGS ON ROUTING LIMITATIONS IN FINE-GRAINED MoE

### 3.1 INSIGHTS FROM PRE-TRAINING PATTERNS

Recent evidence from OLMoE (Muennighoff et al., 2025) and OpenMoE (Xue et al., 2024) suggests that during pre-training, routers converge much earlier than the experts. In particular, OLMoE reports that after only 1% of pre-training, around 60% of the routing decisions have already saturated, while OpenMoE finds that by 40% of pre-training, about 80% of the routing has stabilized. Such early fixation implies that routing decisions are largely determined before the experts have sufficiently matured. Moreover, the widely used balancing loss (Lepikhin et al., 2020), while effective at preventing collapse to just a few experts, may also discourage tokens from concentrating on the most impactful experts. This implies that the current pre-training paradigm for MoE makes a trade-off: *in exchange for training stability, it locks routers into decisions before experts have fully matured.* Combined with balancing constraints, this underutilizes high-potential experts and inflates the role of those that contribute little at inference. This presents an opportunity from a post-training perspective:

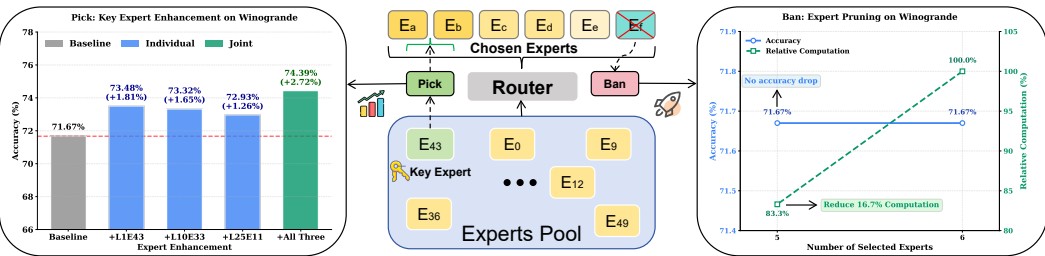

Figure 1: The intuition and empirical findings on expert utilization. Left (**Pick**): forcibly activating key experts (e.g., E43) improves accuracy on Winogrande (Sakaguchi et al., 2021). Right (**Ban**): reducing experts from 6 to 5 cuts 16.7% computation without accuracy loss.

*Can we recover this lost potential—amplifying the influence of the most impactful experts while trimming away redundancy—to both boost performance and accelerate inference?*

### 3.2 EXPERT UTILIZATION INEFFICIENCIES AND OPTIMIZATION POTENTIAL

To validate the inefficiencies in expert utilization and better understand the optimization potential, we conducted two simple yet revealing experiments with the Deepseek-V2-Lite-Chat model (activates 6 experts out of 64). Figure 1 demonstrates how we adjust expert utilization and its effects.

**(1) Underutilization of the most impactful experts:** we examined expert selection frequency on the C4 (Dodge et al., 2021) calibration set, identified the top-3 most frequently selected experts in each layer, and then conducted a grid search: each of these experts was manually added in addition to the original routing (E43 in Figure 1), and the resulting accuracy on Winogrande benchmark was recorded. The left part highlights a few experts contribute disproportionately: enhancing Layer-1 Expert-43 improved accuracy by +1.81%, L10E33 by +1.65%, and L25E11 by +1.26%. When these three were jointly enhanced, accuracy increased by +2.72% (71.67% → 74.39%), while most other high-frequency experts had negligible or noisy effects, indicating that certain impactful experts are insufficiently leveraged in the original routing strategy. We refer to these three experts as **key experts**.

**(2) Computational waste from low-contribution experts:** we directly reduced the number of selected experts per token from 6 to 5 and measured the resulting accuracy. Remarkably, as shown in the right side of Figure 1, this pruning reduced computation by 16.7% and surprisingly preserved full accuracy on Winogrande (71.67%), demonstrating redundancy in the original expert utilization.

Together, these experiments validate the inefficiencies in expert utilization predicted by pre-training analysis, and support the intuition of our proposed methods—Pick and Ban, respectively, which aim to optimize expert selection beyond pre-training constraints for better inference-time performance. This empirical study motivates us to design a unified framework that can pick key experts while banning redundant ones, aiming at improving both accuracy and efficiency.

## 4 PICK

In this section, we begin with an analysis of expert specialization in fine-grained MoE models to verify the existence of domain-specialized experts, which forms the foundation for key experts to effectively function. Building on this analysis, we then introduce our Pick method in two steps: (i) identifying key experts from the set of domain-specialized experts, and (ii) designing strategies to enhance these key experts so that their impact on model performance can be effectively amplified.

### 4.1 EXPERT SPECIALIZATION IN FINE-GRAINED MOE

First, we take a closer look at the degree of expert specialization in fine-grained MoE models. Specifically, we analyze the Qwen3-30B-A3B model and record expert usage frequencies on task-related benchmarks: math tasks (GSM8K + Math-500), code tasks (MBPP (Austin et al., 2021) + Livecodebench), and a general task (GPQA-Diamond). The statistics cover all tokens in both the prefill and decode stages. For each type of task, we plot the per-layer expert usage frequency and show 3 representative layers in Figure 2 (complete results in Appendix A.3).

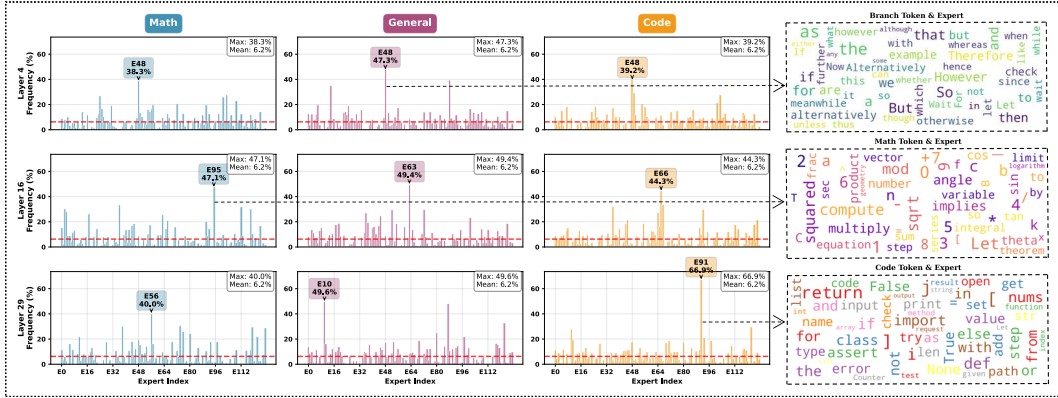

Figure 2: Expert specialization in fine-grained MoE (Qwen3-30B-A3B). Left: expert usage frequency across three tasks (math, general, code). Right: token word clouds for 3 high-frequency experts.

Our analysis reveals that in each layer there exist experts whose usage frequency is significantly higher than average–often up to 6 times higher. Moreover, in most layers these high-frequency experts vary across tasks. For instance, L16E95 is frequently activated only on math tasks, whereas L29E91 is almost exclusively activated on code tasks with an extremely high activation rate of 66.9%. In contrast, a small number of experts such as L4E48 remain highly active across all categories of tasks.

These task-dependent high-frequency experts suggest the existence of domain-specialized experts. To validate this hypothesis, we further examine the functional roles of these experts by analyzing which tokens are most often routed to them. Taking the three aforementioned experts as examples, we visualize the tokens they frequently process using word clouds (after removing empty or semantically meaningless tokens for readability). The results show that the math-specialized expert L16E95 is strongly associated with digits, operators, and math terminology; the code-specialized expert L29E91 is associated with common programming tokens; and the general expert L4E48 is closely tied to branch tokens (Zhong et al., 2025), which play a critical role in chain-of-thought across diverse tasks.

These observations provide empirical evidence that expert specialization in fine-grained MoE models is highly pronounced. Different tasks rely heavily on different domain-specialized experts, while a small subset of general experts remain consistently important across domains.

## 4.2 PICKING KEY EXPERTS FROM DOMAIN-SPECIALIZED EXPERTS

As shown previously, almost every layer contains domain-specialized experts that are activated with notably high frequency. The key issue, however, is whether all of these experts contribute equally to model's performance, or only a small subset is truly essential. To investigate this, we evaluate the importance of each expert by measuring the shift in the logits distribution when it is removed.

Concretely, taking math tasks as an example, we use the domain-specialized experts identified in the previous analysis as candidate experts in each layer. We then run the model on 1000 randomly sampled instances from the MathQA (Amini et al., 2019) dataset. For each sample, we record the output logits of the original model, and then repeat the inference while pruning one candidate expert at a time. The perturbation of the output distribution is quantified using the KL divergence between the probability distributions of the Top-1000 tokens from the original and pruned models (see Appendix A.4 for implementation details and results on other task categories).

The results are summarized in Figure 3. Surprisingly, despite their high activation frequency, most domain-specialized experts cause only negligible shifts in the output distribution when pruned. In contrast, a small number of experts stand out, with three of them inducing particularly large changes. Among these, expert L9E18 is the most striking, leading to a divergence of about 0.16. These findings suggest that only a few domain-specialized experts truly have a decisive influence on the model's behavior, and we hypothesize that they are the key experts we aim to identify.

To test this hypothesis, we construct a failure set consisting of the Math-500 (Lightman et al., 2023) problems that the base model originally answers incorrectly (24 in total). For each candidate expert, we forcibly include it during routing and then re-evaluate accuracy on this failure set. As shown in

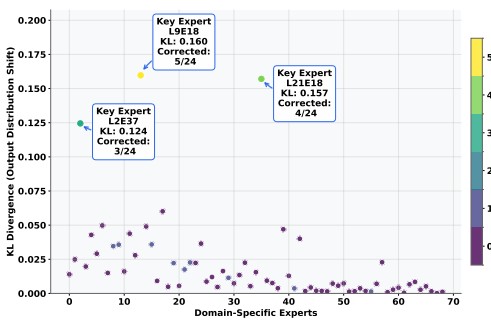 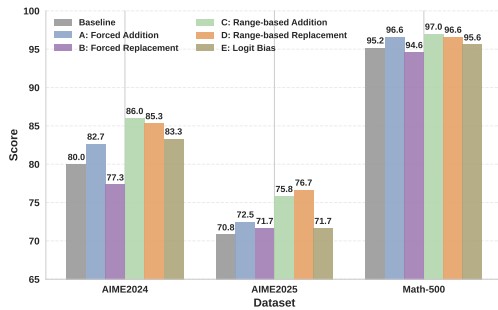

Figure 3: Impact of pruning (measured by KL divergence, left axis) and enhancing (by accuracy gain, right axis) for domain-specialized experts.

Figure 4: Comparison of five designed enhancement methods for key experts, evaluated by accuracy on three widely used math benchmarks.

Figure 3 (right vertical axis), only those experts that induce substantial distribution shifts also yield meaningful accuracy gains when enhanced, whereas others contribute little to no improvement. In particular, expert L9E18, which caused the largest divergence, also increases the number of correct answers by 5 out of 24 when included. This provides strong evidence of the close correspondence between distribution shifts and functional importance, and offers a principled way to distinguish key experts from the many domain-specialized experts.

### 4.3 Enhancing Key Experts

After identifying key experts, the remaining challenge is to determine how to leverage them most effectively. We take math tasks as a running example and design five enhancement strategies:

**A. Forced Addition:** Whenever a key expert is not selected, it is added alongside the chosen experts.

**B. Forced Replacement:** Similar to forced addition, but the key expert replaces the selected expert with the lowest routing weight, maintaining the total number of selected experts.

**C. Range-based Addition:** If a key expert is not selected (not in the top-$k$ candidates) but appears within the router's top-$2k$ candidates, it is forcibly added.

**D. Range-based Replacement:** Under the same condition as range-based addition, the key expert replaces the selected expert with the lowest routing weight.

**E. Logit Bias:** Before routing, a fixed bias is added to the routing logits of key experts. In our implementation, the bias is set to $20\%$ of the average weight among the top-$k$ selected experts.

We evaluate these methods on AIME2024, AIME2025, and Math-500. For the two AIME datasets, results are averaged over five runs to mitigate small-sample variance. Figure 4 summarizes the outcomes. Overall, strategy A brings consistent but modest gains. Strategy B, which forces replacement regardless of probability, is unstable and even reduces accuracy on AIME2024 and Math-500, indicating that overly aggressive intervention can be harmful. By contrast, strategies C and D—enhancing only when the key expert is already close to the router's top candidates—deliver the strongest improvements across all benchmarks. The difference between C and D is minor, but since D avoids expanding the number of active experts, it is preferable in practice and used in our experiments. Strategy E shows less stable performance, as the bias magnitude is sensitive and difficult to tune.

## 5 BAN

Our dynamic expert pruning method is motivated by two key observations: 1) Different layers in an MoE model exhibit highly diverse sensitivity to expert pruning and thus should be treated differently. 2) During reasoning, the token-wise routing distributions also vary significantly: when most of the routing weight is concentrated on a few top experts, expert pruning causes minimal loss; in contrast, when the weights are more evenly distributed, expert pruning becomes more harmful.

In the following, we first use Qwen3-30B-A3B as an example to illustrate the differences in layer sensitivity and token sensitivity, and then introduce our proposed dynamic pruning method.

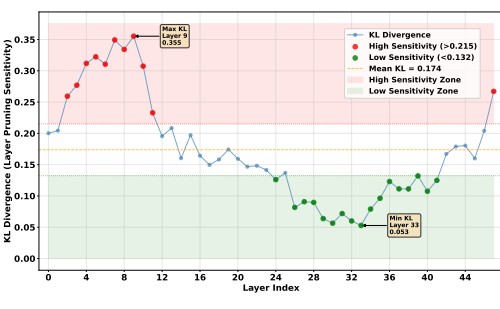

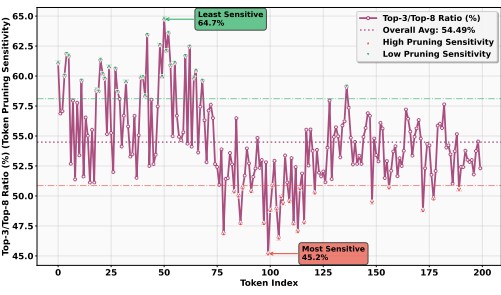

(a) Layer-wise pruning sensitivity.

(b) Token-wise pruning sensitivity.

Figure 5: Sensitivity analysis of MoE experts: (a) layer-wise and (b) token-wise. Both dimensions exhibit large variance, motivating a dynamic pruning strategy.

**Layer Sensitivity.** We measure layer-wise pruning sensitivity by reducing the number of selected experts from $k = 8$ to $k = 3$ in each layer, and computing the Top-1000 KL divergence between the output logits of the original and pruned models on MathQA calibration set. The results (Figure 5a) reveal substantial differences across layers: while some layers, especially at the begin and end of the model, are highly sensitive to pruning, many middle layers are much more robust. This large variance highlights the necessity of treating different layers differently when designing pruning strategies.

**Token Sensitivity.** We further analyze token-level pruning sensitivity in long reasoning processes by examining how routing weights are distributed across experts. To quantify this, we compute the ratio between the cumulative routing weights of the top-3 and top-8 experts, averaged across all layers. Figure 5b plots these ratios for the first 200 generated tokens in a reasoning process from one GPQA-Diamond sample. The results reveal substantial variability across tokens: some tokens concentrate most of their weights on a few experts and are thus robust to expert pruning, whereas others distribute weights more evenly and are therefore more sensitive.

**Dynamic pruning.** Before combining layer- and token-level information, we normalize their sensitivity scores to the range $[0, 1]$. Here, $W_l$ denotes the KL divergence of layer $l$ when reducing expert selection from $k = 8$ to $k = 3$. $W_{\min}$ and $W_{\max}$ are defined as the divergences of the least-sensitive and most-sensitive layers, respectively. For tokens, $R_i$ represents the top-3/top-8 routing weight ratio of token $i$ at a specific layer during inference, with $R_{\min}$ and $R_{\max}$ similarly estimated from the calibration set. The normalized sensitivity scores are then defined as:

$$L'_l = (W_l - W_{\min})/(W_{\max} - W_{\min}), \qquad T'_i = (R_{\max} - R_i)/(R_{\max} - R_{\min}).$$

The combined sensitivity score for token $i$ at layer $l$ is then:

$$S_{i,l} = \lambda \cdot \tfrac{1}{2}\left(L'_l + T'_i\right), \quad \lambda \in (0, 1)$$

where we simply average layer- and token-level sensitivities to balance their contributions, and use $\lambda$ to control the aggressiveness of pruning. The dynamic number of selected experts is determined as:

$$K_{i,l} = \left\lfloor K_{\min} + (K_{\text{base}} - K_{\min}) \cdot S_{i,l} \right\rfloor.$$

Here $K_{\text{base}} = 8$ is the default number of experts, and $K_{\min} = 3$ serves as a safeguard to prevent selecting too few experts: from empirical inspection, we found that reducing the number of selected experts below this threshold leads to a complete collapse of the model. Details of the experimental validation for the setting of $K_{\min}$ and the choice of $\lambda$ are provided in the Appendix A.5.

## 6 EXPERIMENTS

In this section, we first introduce the experimental setup, including models, datasets, and implementation details. We then separately compare our proposed **Pick** and **Ban** approaches with existing methods, highlighting their respective strengths. After that, we demonstrate how combining **Ban&Pick** achieves both improved performance and faster inference. Finally, we examine the influence of key experts on tasks from other domains, offering insights for future research directions.

## 6.1 Experiment Setup

**Models and Datasets**    We conduct experiments on four fine-grained MoE models from two different model families, covering a wide range of parameter scales: DeepSeek-V2-Lite-Chat (16B), DeepSeek-V2.5-1210 (236B), Qwen3-30B-A3B, and Qwen3-235B-A22B. Our evaluation spans three categories of tasks using five commonly adopted datasets: AIME2024 (with 5 runs) (Patel et al., 2024) and Math-500 for mathematical reasoning, GPQA-Diamond (Rein et al., 2023) for general knowledge reasoning, and LiveCodeBench (V1) (Naman Jain, 2024) together with HumanEval+ (Liu et al., 2023) for code generation. For DeepSeek-V2-Lite-Chat, we replace AIME2024 with GSM8K, as AIME2024 is overly difficult for this smaller model. We use OpenCompass (Contributors, 2023) as evaluation toolkit. The detailed generation hyperparameters and prompts are provided in the Appendix A.6.

Table 1: Comparison of **Pick** against three baselines across four models. Reported accuracies cover multiple benchmarks, with GSM8K (Cobbe et al., 2021) only for DeepSeek-V2-Lite-Chat.

| Models | Methods | GSM8K/AIME24 | Math500 | GPQA | LiveCodeBench | HumanEval+ |
|---|---|---|---|---|---|---|
| Deepseek-V2-Lite-Chat | Baseline | 66.34 | 26.40 | 24.75 | 9.25 | 50.61 |
| | Dynamic | 66.56 | 26.80 | 24.75 | 9.00 | 50.61 |
| | Tips | 66.87 | 26.60 | 25.25 | 9.50 | 51.22 |
| | Pick | **67.48** | **28.00** | **25.76** | **10.50** | **51.83** |
| Deepseek-v2.5-1210 | Baseline | 23.33 | 81.00 | 47.47 | 60.75 | 83.54 |
| | Dynamic | 24.00 | 81.20 | 46.97 | 61.00 | 83.54 |
| | Tips | 24.67 | 80.60 | 47.98 | 61.25 | 84.15 |
| | Pick | **27.34** | **82.80** | **49.49** | **62.75** | **84.76** |
| Qwen3-30B-A3B | Baseline | 80.67 | 95.20 | 65.66 | 91.75 | 71.34 |
| | Dynamic | 82.66 | 94.80 | 66.16 | 92.00 | 71.95 |
| | Rice | 83.33 | 95.60 | 67.68 | 93.50 | 72.56 |
| | Pick | **85.33** | **96.60** | **69.19** | **94.25** | **75.61** |
| Qwen3-235B-A22B | Baseline | 84.67 | 96.00 | 71.21 | 92.75 | 78.05 |
| | Dynamic | 85.33 | 96.00 | 71.72 | 93.00 | 78.66 |
| | Rice | 86.67 | 96.20 | 72.22 | 93.00 | 79.88 |
| | Pick | **88.00** | **96.80** | **74.24** | **94.50** | **81.10** |

**Implementation Details**    For the Pick method, we follow the procedures described in Section 4 to identify key experts for each of the three task categories–math, code and general reasoning–and apply targeted enhancements to them. We then evaluate the resulting models on their corresponding task types. For the Ban method, we set $K_{\min} = 3$ and $\lambda = 0.7$ for all four models, aiming to achieve a balanced trade-off between accuracy and speedup (further discussed in Appendix A.5). When combining Pick and Ban, we note that in layers containing key experts, Ban has already pruned redundant experts, so directly applying the range-based replacement (strategy D) leads to noticeable performance degradation. To address this, we instead adopt the range-based addition (strategy C).

## 6.2 Results of Pick

We compare our proposed **Pick** method against three representative baselines, covering both routing-strategy adjustments and test-time scaling approaches (see Appendix A.7 for reproducibility details):
(1) **Dynamic Routing.** (Huang et al., 2024a) adaptively allocates experts according to routing weights, assigning more experts to harder tokens and fewer to easier ones, realized by accumulating experts until a threshold is reached. We set thresholds $\{0.7, 0.8, 0.9\}$ and report the best result.
(2) **Tip.** (Wang et al., 2025b) penalizes tokens that typically indicate switching paths, thereby encouraging the model to continue deepening its current line of reasoning.
(3) **Rice.** (Wang et al., 2025a) leverages explicit `<think>` tokens to identify cognitive experts that specialize in reasoning. During inference, the routing weights of these experts are multiplied by a constant factor to amplify their influence. Since RICE relies on explicit `<think>` tokens and is therefore inapplicable to the two DeepSeek models, and it has reported better results than Tip on Qwen3 series, we adopt Tip as the baseline for DeepSeek and Rice for Qwen3 in our experiments.

As shown in Table 1, **Pick** achieves consistent and notable improvements across four models on five datasets, outperforming all three baselines. The gains are especially pronounced on the Qwen3 series, which we attribute to their stronger expert specialization. Remarkably, on the large-scale Qwen3-235B-A22B, whose performance is already comparable to leading closed-source models,

Pick further boosts accuracy by 3.33%, 3.03%, and 3.05% on AIME2024, GPQA, and HumanEval+, respectively. These results highlight the potential of leveraging key experts to further improve MoE model performance. Meanwhile, on DeepSeek-v2.5-1210, Pick achieves an average improvement of 1.97% across five datasets, demonstrating strong generalization capability.

## 6.3 RESULTS OF BAN

Table 2: Comparison of **Ban** against DES and ODP across four models. We report average experts per token (Avg-Topk), accuracy, and inference speedup under the vLLM (0.9.1) backend.

| Models | Methods | Avg-Topk | GSM8K/AIME24 | Math500 | GPQA | LiveCodeBench | HumanEval+ | Speedup |
|---|---|---|---|---|---|---|---|---|
| Deepseek-V2-Lite-Chat | Baseline | 6.00 | 66.34 | 26.40 | 24.75 | 9.25 | 50.61 | 1.00 |
| | DES | 4.18 | 63.76 | 21.40 | 22.22 | 7.25 | 45.73 | 1.20 |
| | ODP | 4.22 | 64.76 | 23.20 | 23.23 | 7.75 | 46.95 | 1.19 |
| | Ban | 4.02 | **66.11** | **25.00** | **24.24** | **8.50** | **48.17** | **1.22** |
| Deepseek-v2.5-1210 | Baseline | 6.00 | 23.33 | 81.00 | 47.47 | 60.75 | 83.54 | 1.00 |
| | DES | 4.21 | 22.66 | 80.20 | 39.90 | 58.00 | 79.27 | 1.21 |
| | ODP | 4.29 | 23.33 | 80.20 | 43.43 | 59.25 | 81.71 | 1.20 |
| | Ban | 3.97 | **23.33** | **80.60** | **45.96** | **60.25** | **82.93** | **1.24** |
| Qwen3-30B-A3B | Baseline | 8.00 | 80.67 | 95.20 | 65.66 | 91.75 | 71.34 | 1.00 |
| | DES | 4.94 | 73.33 | 92.60 | 56.06 | 84.50 | 65.24 | 1.25 |
| | ODP | 4.98 | 74.67 | 93.20 | 60.10 | 86.00 | 67.07 | 1.24 |
| | Ban | 4.82 | **80.00** | **94.60** | **64.65** | **90.00** | **70.12** | **1.25** |
| Qwen3-235B-A22B | Baseline | 8.00 | 84.67 | 96.00 | 71.21 | 92.75 | 78.05 | 1.00 |
| | DES | 4.89 | 68.00 | 94.80 | 62.12 | 90.75 | 70.12 | 1.26 |
| | ODP | 4.91 | 73.33 | 95.20 | 66.16 | 91.00 | 73.78 | 1.26 |
| | Ban | 4.77 | **83.33** | **95.40** | **69.70** | **91.25** | **75.61** | **1.27** |

We compare our proposed **Ban** method against two mainstream dynamic expert-pruning approaches: (1) **DES.** (Lu et al., 2024) (Dynamic Expert Skip) dynamically prunes low-weight experts by observing that, for easy tokens, the contribution of lower-ranked experts is often negligible. Concretely, it decides whether to drop the secondary expert based on the ratio between expert routing weights. (2) **ODP.** (Huang et al., 2024b) (Online Dynamic Pruning) extends DES by adding a key-token protection mechanism. Tokens that receive high attention scores are exempted from pruning.

In terms of accuracy, as shown in Table 2, both DES and ODP suffer from severe degradation on these challenging tasks, with drops exceeding 5% on certain datasets. In contrast, Ban limits the accuracy loss to within 1.5% on most models and datasets, except HumanEval+, which we found sensitive to pruning, and even within 1% on more than half of the results. Notably, on Qwen3-30B-A3B, Ban reduces accuracy on AIME2024 by only 0.67%–equivalent to misanswering just one additional problem out of five evaluations–and incurs merely 1.01% degradation on GPQA.

In addition to accuracy, we record the average number of selected experts per token and the relative speedup—computed from the total inference time across all datasets—under the vLLM backend to comprehensively evaluate inference efficiency. **Ban** restricts the average number of selected experts to around 4 for DeepSeek models, leading to more than $1.2\times$ actual speedup. On Qwen3 models, Ban keeps the average expert count below 5, achieving about $1.25\times$ speedup. These results demonstrate the strong ability of Ban to preserve model performance while delivering substantial efficiency gains.

## 6.4 RESULTS OF BAN&PICK

Table 3: Results of combining **Ban&Pick** across four models. We report average experts per token, accuracy, and inference speedup under vLLM. Relative changes from baseline are annotated in gray.

| Models | Methods | Avg-Topk | GSM8K/AIME24 | Math500 | GPQA | LiveCodeBench | HumanEval+ | Speedup |
|---|---|---|---|---|---|---|---|---|
| Deepseek-V2-Lite-Chat | Baseline | 6.00 | 66.34 | 26.40 | 24.75 | 9.25 | 50.61 | 1.00 |
| | Ban&Pick | 4.03 | 66.94 ↑0.60 | 26.80 ↑0.40 | 25.25 ↑0.50 | 9.50 ↑0.25 | 50.00 ↓0.61 | 1.22 |
| Deepseek-v2.5-1210 | Baseline | 6.00 | 23.33 | 81.00 | 47.47 | 60.75 | 83.54 | 1.00 |
| | Ban&Pick | 3.99 | 26.00 ↑2.67 | 81.60 ↑0.60 | 47.98 ↑0.51 | 61.25 ↑0.50 | 84.15 ↑0.61 | 1.23 |
| Qwen3-30B-A3B | Baseline | 8.00 | 80.67 | 95.20 | 65.66 | 91.75 | 71.34 | 1.00 |
| | Ban&Pick | 4.83 | 84.66 ↑3.99 | 95.80 ↑0.60 | 68.18 ↑2.52 | 92.85 ↑1.10 | 72.56 ↑1.22 | 1.25 |
| Qwen3-235B-A22B | Baseline | 8.00 | 84.67 | 96.00 | 71.21 | 92.75 | 78.05 | 1.00 |
| | Ban&Pick | 4.79 | 86.67 ↑2.00 | 96.00 ↑0.00 | 72.73 ↑1.52 | 93.25 ↑0.50 | 78.66 ↑0.61 | 1.26 |

By combining **Ban** and **Pick**, our unified framework leverages smarter routing strategies to simultaneously deliver free accuracy gains and inference acceleration across models and datasets,

demonstrating strong practicality. As shown in Table 3, on Qwen3-30B-A3B, Ban&Pick achieves an average accuracy improvement of 1.99% together with a $1.25\times$ speedup, while on the large-scale Qwen3-235B-A22B, it yields an average accuracy gain of 1.33% and a $1.26\times$ speedup. As two plug-and-play modules, Ban and Pick share the common goal of utilizing smarter routing to release the potential of fine-grained MoE, while offering flexible choices: higher performance (**Pick**), faster inference (**Ban**), or a balanced combination of both improvements (**Ban&Pick**).

# 7 INTERACTION AMONG KEY EXPERTS ACROSS TASKS

In this section, we further investigate the scenario where all identified key experts across math, general reasoning, and code are enhanced simultaneously. We first summarize the enhanced experts for each task type in Table 4. Then, using Qwen3-30B-A3B as a case study, we report in Table 5 the performance when enhancing key experts for a single domain, for domain pairs (Math+General, Math+Code, General+Code), and for all three domains together. This allows us to examine whether experts from different domains complement or interfere with each other when activated jointly, providing a more comprehensive understanding of cross-domain interactions.

Table 4: Enhanced key experts (in the format (layer, expert) ) for different domains across models.

| Model | Math Experts | General Experts | Code Experts |
|---|---|---|---|
| Qwen3-30B-A3B | (2,37),(9,18),(21,18) | (5,19),(7,114),(13,105) | (6,40),(18,40) |
| Qwen3-235B-A22B | (11,121),(31,121),(51,94) | (13,57),(18,91),(62,101) | (7,56),(62,48),(82,48) |
| DeepSeek-V2-Lite-Chat | (5,14),(9,56),(17,25) | (4,36),(7,31),(14,22) | (8,1),(12,28) |
| DeepSeek-v2.5-1210 | (6,58),(32,5) | (9,22),(16,66) | (1,129),(22,36),(35,13) |

Table 5: Task accuracies (%) on Qwen3-30B-A3B under different combinations of enhanced experts. A ✓ indicates that the corresponding domain experts are enhanced, while × means not enhanced.

| Math | General | Code | AIME2024 | MATH-500 | GPQA-Diamond | LiveCodeBench | HumanEval+ |
|---|---|---|---|---|---|---|---|
| × | × | × | 80.67 | 95.20 | 65.66 | 91.75 | 71.34 |
| ✓ | × | × | 85.33 ↑4.66 | 96.60 ↑1.40 | 65.15 ↓0.51 | 91.50 ↓0.25 | 71.95 ↑0.61 |
| × | ✓ | × | 82.00 ↑1.33 | 95.20 ↑0.00 | 69.19 ↑3.53 | 91.75 ↑0.00 | 72.56 ↑1.22 |
| × | × | ✓ | 81.33 ↑0.66 | 95.00 ↓0.20 | 65.66 ↑0.00 | 94.25 ↑2.50 | 75.61 ↑4.27 |
| ✓ | ✓ | × | 84.67 ↑4.00 | 96.20 ↑1.00 | 67.68 ↑2.02 | 92.00 ↑0.25 | 71.34 ↑0.00 |
| ✓ | × | ✓ | 84.67 ↑4.00 | 96.00 ↑0.80 | 66.16 ↑0.50 | 93.50 ↑1.75 | 74.39 ↑3.05 |
| × | ✓ | ✓ | 80.67 ↑0.00 | 94.80 ↓0.40 | 68.18 ↑2.52 | 92.75 ↑1.00 | 75.00 ↑3.66 |
| ✓ | ✓ | ✓ | 84.00 ↑3.33 | 96.00 ↑0.80 | 67.68 ↑2.02 | 93.00 ↑1.25 | 74.39 ↑3.05 |

From Table 5, we observe that enhancing multiple domains simultaneously still brings clear improvements. For instance, when both Math and General experts are enhanced, AIME2024 increases by +4.0% and GPQA by +2.02%. However, these gains are slightly smaller than enhancing each domain individually ($-0.66\%$ and $-1.51\%$, respectively). When all three domains are enhanced together, the model continues to outperform the baseline—+3.33% on AIME2024, +2.02% on GPQA, and +3.05% on HumanEval+—though the gains are somewhat reduced compared to the best single-domain setting. These results indicate that while domain-specific key experts do interfere to some extent, their combined enhancement remains clearly beneficial, confirming their crucial role.

# 8 WHY KEY EXPERTS ARE UNDERUTILIZED IN FINE-GRAINED MOE

In this section, we provide a theoretical analysis of why fine-grained MoE routers fail to sufficiently select some key experts. Our central claim is **premature routing** and **auxiliary balancing loss** jointly lead to under-utilization of key experts, even though they become highly specialized and impactful by the end of training. Below, we first review the balancing loss and its behavior in fine-grained MoE, and then, based on these two aspects, explain the resulting under-utilization of key experts.

**Background: auxiliary balancing loss and its effect.** In early MoE architectures, each layer typically contains only a small number of experts, and each token is routed to 1–2 experts. Without

additional constraints, training tends to collapse onto a few experts that are activated very frequently, while the remaining experts receive too few updates and effectively degrade. To mitigate this, prior work (Lepikhin et al., 2020; Fedus et al., 2022) introduced an auxiliary balancing loss in each layer that explicitly regularizes the router toward more even usage. A simplified form is

$$\mathcal{L}_{\text{expert}} = \alpha \sum_i f_i \cdot p_i, \tag{1}$$

where $f_i$ is the actual activation frequency of expert $i$, $p_i$ is the average routing score (mean softmax probability over tokens) assigned to expert $i$, and $\alpha$ is a balancing coefficient. Intuitively, when an expert receives both high routing probability and high actual usage, the contribution $f_i p_i$ becomes large and is penalized. This mechanism is effective at preventing collapse and yields balanced loads.

However, as also noted in follow-up work like Wang et al. (2024), the auxiliary balancing loss conflicts with one of the core goals of MoE: expert specialization. Ideally, the MoE model should be encouraged to develop specialized experts that are responsible for different tasks. But the balancing loss discourages this when such specialization becomes pronounced. As a result, there is an inherent conflict between expert specialization and training stability that is widely present in MoE training.

**Fine-grained MoE: partial relief but persistent constraints.** Fine-grained MoE designs increase the number of experts per layer dramatically. As a result, the summation in equation 1 spans many more experts, so the influence of each individual expert on the overall term becomes smaller, which allows some experts to have noticeably higher frequencies while keeping training relatively stable, as long as these frequencies are not excessively large. Leveraging this property, fine-grained MoE finds a more effective balance between training stability and expert specialization: it maintains stable training while still enabling a non-trivial degree of expert specialization to emerge.

These effects are consistent with our analysis in Section 4.1. In Qwen3-30B-A3B, we observe clear domain specialization: different tasks rely heavily on different subsets of experts. However, even for strongly domain-specialized experts, their activation frequencies rarely exceed $\sim 50\%$ for a given domain, indicating that auxiliary balancing loss still caps how dominant single expert can become.

**Under-utilization of key experts.** We now explain how premature routing and auxiliary balancing loss jointly lead to under-utilization of key experts. Conceptually, this process can be divided into two stages during pre-training. In the early stage of fine-grained MoE pre-training, before routing prematurely converges, the router already learns a relatively stable mapping from task types to experts: different domains (e.g., math, code, general) are routed to different subsets of experts, and expert specialization starts to emerge. At the same time, however, the auxiliary balancing loss limits how high the activation frequency of any single expert can become for a given domain, preventing overly concentrated routing even when some experts are more suitable for that domain.

When routing becomes premature, the routing-related loss has essentially converged and the routing pattern becomes stable: the selection probabilities for each expert no longer adapt significantly. For the remainder (which constitutes the majority) of pre-training, the frequently selected experts for each domain continue to receive large amounts of domain-specific data, so their parameters are updated heavily and they become increasingly specialized for that domain. **As a result, some of these experts evolve into key experts that are particularly important for certain tasks, but their activation frequencies remain capped by the earlier, balance-constrained routing decisions instead of increasing in proportion to their improved competence.** Our Pick method is designed precisely to compensate for this gap at inference time, by manually enhancing the usage of these key experts so that their activation frequency better matches their actual specialization and impact on performance.

## 9 CONCLUSION AND IMPLICATIONS FOR FUTURE WORK.

In this study, we reveal that in fine-grained MoE models, relatively balanced routing strategy underutilizes a small set of key experts which have a disproportionately large impact on task performance, and fixed expert activation introduces substantial redundancy. By selectively increasing the utilization of key experts and dynamically pruning redundant ones, we release the potential of routing and provide a novel and effective framework that improves both accuracy and efficiency. For future work, we suggest exploring the underlying mechanisms of how key experts enhance model performance, and investigating more sophisticated enhancement strategies beyond uniform boosting.

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

# A APPENDIX

## A.1 LLM USAGE

We acknowledge the use of Large Language Models (LLMs) to improve the clarity and quality of the written presentation in this work.

## A.2 MIXTURE-OF-EXPERTS

The Sparsely-Gated MoE (Jacobs et al., 1991; Jordan & Jacobs, 1994) first introduced the idea of using a trainable gating network to activate only a small subset of experts, typically through a Top-K routing scheme. Subsequent studies (DeepSeek-AI et al., 2024; Yang et al., 2025) built upon this framework, scaling MoE models to much larger sizes and demonstrating that sparse expert activation can deliver strong efficiency and performance gains. As these models expanded, the routing mechanism became a critical factor in determining overall effectiveness.

The Sparsely-Gated MoE are based on a transformer architecture, but the FeedForward Network (FFN) sub-layers of traditional dense models are replaced with MoE layers, each containing $N$ experts. For each input token $x$, the router computes routing logits $r = \{r_0, \cdots, r_{N-1}\}$ and expert selection scores $s = \text{Softmax}(r)$. The top-$K$ experts are selected based on $s$, and their outputs $E_{e_j}(x)$ are combined as a weighted sum, with normalized weights:

$$z = \sum_{j=0}^{K-1} \frac{s_{e_j}}{\sum_{i=0}^{K-1} s_{e_i}} \cdot E_{e_j}(x). \tag{2}$$

Here, $E_{e_j}(x)$ represents the output of the $j$-th selected expert for the input token $x$. Notably, different MoE architectures handle expert weight combination differently. Qwen3 employs the weight renormalization mechanism shown in Equation 2, where the selected expert weights are normalized to sum to 1. In contrast, DeepSeek-v2 uses direct weight scaling without renormalization. Moreover, DeepSeek-v2 incorporates grouped expert selection, where experts are partitioned into groups to improve specialization. Detailed settings and implementation can be checked in their papers (DeepSeek-AI et al., 2024).

Despite such architectural differences, both lines of work–and many other recent systems–adopt a fine-grained MoE design: each layer hosts a large number of small, independently initialized experts. This design enhances expert specialization but simultaneously increases the difficulty of ensuring

Table 6: Specifications of fine-grained MoE models used in our experiments.

| Model | Experts | Top-K | Shared-Experts | Layers | Parameters |
|---|---|---|---|---|---|
| Deepseek-V2-Lite-Chat | 64 | 6 | 2 | 27 | 16B |
| Deepseek-v2.5-1210 | 160 | 6 | 2 | 60 | 236B |
| Qwen3-30B-A3B | 128 | 8 | 0 | 96 | 30B |
| Qwen3-235B-A22B | 128 | 8 | 0 | 48 | 235B |

efficient routing. In our experimental evaluation, we examine four representative fine-grained MoE models, whose detailed architectural configurations and parameter settings are summarized in Table 6. Note that in the DeepSeek-v2 architecture design (including Deepseek-V2-Lite-Chat and Deepseek-v2.5-1210), the first layer is a dense layer for training stability, while the remaining layers are MoE layers. In contrast, the Qwen3 series uses MoE layers throughout all layers.

## A.3 COMPLETE RESULTS ON EXPERT SPECIALIZATION

In Section 4.1, we analyzed the specialization of experts in fine-grained MoE models using Qwen3-30B-A3B as a case study, and presented representative figures for a subset of layers. Here, we provide the complete visualization of expert usage frequencies across all layers of the model. Results are shown for three representative task types—math, code, and general. For each layer, the bar charts display the relative usage frequency of each expert, aggregated over all tokens in both the prefill and decode stages. Figures are grouped every 12 layers for readability. These results (Figures 6–9) confirm that different task types consistently activate distinct subsets of experts, highlighting clear patterns of domain specialization.

## A.4 COMPLETE RESULTS ON IDENTIFYING KEY EXPERTS

In Section 4.2, we introduced our method for identifying key experts by measuring the KL divergence between the output distributions of the original model and variants where individual domain-specific experts are pruned. There, we focused on Qwen3-30B-A3B with math tasks as a case study, to illustrate how only a small subset of high-frequency experts exert a decisive influence on model predictions. Here, we provide the complete results across **four different models** and **three task types** (math, general, and code). For each model, we report the KL divergence values of candidate experts when pruned, visualized as bar charts.

To prepare for this analysis, we first collect the **domain-specialized experts** following the procedure in Section 4.1. For each task type, we record the expert usage frequencies of the model on a calibration set. For the Qwen3 series, we define domain-specialized experts as those whose usage frequency exceeds six times the expected frequency under perfectly uniform routing. For the DeepSeek series, where the distribution of expert usage is less sharply peaked, we instead adopt a relaxed threshold of three times the uniform baseline. The experts identified in this way serve as the candidates for the KL divergence experiments.

We then construct the calibration sets used to measure KL divergence: MathQA for math, ARC-Challenge (Clark et al., 2018) for general, and the college-computer-science-compilers subset of MMLU (Hendrycks et al., 2021) for code. We mainly select datasets where only the first token of the answer is required as output, because this simplifies evaluation while still providing an accurate measure of the KL divergence.

Following prior work (Xing Hu & Zhou, 2025), we note that the model's output logits typically follow a long-tail distribution, where only the top-ranked tokens carry substantial probability mass. Therefore, when computing KL divergence, we restrict attention to the **Top-1000 tokens**. Concretely, let $p$ denote the probability distribution obtained from the original model, and $q$ the distribution when a candidate expert is pruned. Denote by $\mathcal{I}_{1000}$ the index set of the top-1000 tokens under $p$. The restricted distributions are defined as

$$p'(i) = \frac{p(i)}{\sum_{j \in \mathcal{I}_{1000}} p(j)}, \quad q'(i) = \frac{q(i)}{\sum_{j \in \mathcal{I}_{1000}} q(j)}, \quad i \in \mathcal{I}_{1000}.$$

The KL divergence used in our analysis is then

$$D_{\mathrm{KL}}(p' \parallel q') = \sum_{i \in \mathcal{I}_{1000}} p'(i) \log \frac{p'(i)}{q'(i)}.$$

This definition ensures that the comparison focuses on the most influential tokens, while ignoring the long tail of near-zero probabilities. The complete results for all models and task types are shown in Figures 10–13. Consistent with the case study in the main text, we observe across different models and domains that only a small subset of high-frequency experts causes substantial divergence, while the majority of experts induce negligible changes.

## A.5 Hyperparameter Choices in Dynamic Pruning

As introduced in Section 5, our dynamic pruning strategy involves two quantities: $K_{\min}$ and $\lambda$. The value of $K_{\min}$ is determined empirically to serve as a safeguard to prevent selecting too few experts In contrast, $\lambda$ acts as the single knob controlling pruning aggressiveness and presents a trade-off between efficiency and accuracy. We discuss its effect through detailed experiments below.

**Setting of $K_{\min}$.** In the main text, we noted that reducing the number of selected experts below a certain threshold causes the model to collapse entirely. Therefore, $K_{\min}$ must guarantee that even when both layer- and token-level sensitivities are low, the number of selected experts never falls below this safe lower bound.

To study this effect, we take Qwen3-30B-A3B as an example and directly enforce the number of experts per token to be fixed between 2 and 7 (with $K_{\mathrm{base}} = 8$ as the default). Table 7 reports the performance on five benchmarks. For efficiency, results on AIME2024 (except the baseline) are measured over a single run.

| TopK | AIME2024 | Math500 | GPQA | Livecodebench | HumanEval+ |
|------|----------|---------|-------|---------------|------------|
| 8 | 80.67 | 95.2 | 65.66 | 91.75 | 71.34 |
| 7 | 80.00 | 95.0 | 65.66 | 91.25 | 71.95 |
| 6 | 76.67 | 93.2 | 63.64 | 89.75 | 69.51 |
| 5 | 76.67 | 88.2 | 59.60 | 87.00 | 66.46 |
| 4 | 63.33 | 78.4 | 54.55 | 74.75 | 59.76 |
| 3 | 40.00 | 55.8 | 44.95 | 57.50 | 43.29 |
| 2 | 16.67 | 19.6 | 21.72 | 25.50 | 17.07 |

Table 7: Performance of Qwen3-30B-A3B under different fixed numbers of selected experts.

We observe that removing just one expert (TopK=7) leads to almost no performance degradation, consistent with the expert pruning observations reported in Section 3 in the main text. However, once two or more experts are pruned, noticeable drops appear across all benchmarks. The most critical boundary occurs at TopK=3: performance collapses when fewer than three experts are selected. Therefore, we set $K_{\min} = 3$ to prevent the model from falling below this threshold. On the DeepSeek series, we observed a similar phenomenon, further reinforcing this choice.

**Choice of $\lambda$.** The hyperparameter $\lambda$ controls the trade-off between accuracy preservation and inference speedup: larger values of $\lambda$ bias the pruning strategy toward retaining more experts (thus preserving accuracy), while smaller values encourage more aggressive pruning (thus improving throughput). To study this trade-off, we take Qwen3-30B-A3B as an example and vary $\lambda$ from $0.5$ to $0.9$ in increments of $0.1$. For each setting, we evaluate accuracy on the same five benchmarks as in the main experiments and measure the inference speedup using `vLLM` as the serving backend. The speedup is calculated as the ratio between the total inference time across all datasets and the baseline time without pruning. All results are reported from a single run.

From Table 8, we observe that when $\lambda$ is below $0.7$, the model achieves higher speedups but suffers noticeable accuracy degradation compared with $\lambda = 0.7$. When $\lambda$ exceeds $0.7$, accuracy improves slightly, but the gains are marginal relative to the loss in efficiency. Considering this trade-off, we set $\lambda = 0.7$ in our main experiments and report results under this configuration. Nevertheless, $\lambda$ remains

| $\lambda$ | Avg TopK | AIME2024 | Math500 | GPQA | Livecodebench | HumanEval+ | Speedup |
|-----------|----------|----------|---------|------|---------------|------------|---------|
|           | 8        | 80.67    | 95.2    | 65.66 | 91.75        | 71.34      | 1.00    |
| 0.5       | 4.33     | 73.33    | 93.8    | 59.60 | 86.50        | 65.85      | 1.30    |
| 0.6       | 4.61     | 76.67    | 94.0    | 63.64 | 89.25        | 68.90      | 1.26    |
| 0.7       | 4.82     | 80.00    | 94.6    | 64.65 | 90.00        | 70.12      | 1.25    |
| 0.8       | 5.09     | 80.00    | 94.0    | 64.65 | 90.75        | 70.12      | 1.22    |
| 0.9       | 5.32     | 80.00    | 94.8    | 64.14 | 90.50        | 70.73      | 1.20    |

Table 8: Accuracy and inference speedup of Qwen3-30B-A3B under different values of $\lambda$.

a tunable knob that allows users to flexibly balance accuracy preservation and inference acceleration according to their own application needs.

## A.6 EVALUATION DETAILS

### A.6.1 DATASET EVALUATION.

For **AIME2024**, due to the small sample size, we follow the common practice in technical reports by evaluating each method five times and reporting the average. For other datasets, constrained by time, we conduct only a single evaluation. On the **GPQA-Diamond** dataset, to further reduce errors caused by misaligned answer formats, we adopt an LLM-judge in addition to the base evaluator: specifically, we deploy a `Qwen2.5-72B-Instruct` model (Qwen et al., 2025) with `vLLM` (Kwon et al., 2023) as the judge model. For all other datasets, we rely on the default base evaluators.

### A.6.2 GENERATION HYPERPARAMETERS.

During generation, for all four models used in our experiments, we adopt the recommended hyperparameters provided in their papers or Model Cards. The configuration is summarized in Table 9.

Table 9: Generation hyperparameters for the evaluated models.

| Model | Top-k | Top-p | Temperature | Max Output Length |
|-------|-------|-------|-------------|-------------------|
| DeepSeek-V2-Lite-Chat | None | 0.95 | 0.3 | 8192 |
| DeepSeek-V2.5-1210 | 1 | 0.90 | 0.3 | 8192 |
| Qwen3-30B-A3B | 20 | 0.95 | 0.6 | 32768 |
| Qwen3-235B-A22B | 20 | 0.95 | 0.6 | 32768 |

### A.6.3 PROMPT

In this section, we list the prompts used for different datasets in our experiments.

**AIME2024.**

```
{question}
Remember to put your final answer within \boxed{}.
```

**MATH-500.**

```
{problem}
Remember to put your final answer within \boxed{}.
```

**GPQA-Diamond.**

```
Answer the following multiple choice question. The last line of your
    response should be of the following format: 'ANSWER: $LETTER'
    (without quotes) where LETTER is one of ABCD. Think step by step
    before answering.
```

```
{question}
A) {A}
B) {B}
C) {C}
D) {D}
```

**LiveCodeBench.**

```
You are an expert Python programmer. You will be given a question
    (problem specification) and will generate a correct Python program
    that matches the specification and passes all tests. You will NOT
    return anything except for the program.
{question}
```

**HumanEval+.**

```
Complete the following python code:
{prompt}
```

**GSM8K.**

```
{question}
Please put your final answer within \boxed{}.
```

## A.7 DETAILS OF COMPARED METHODS AND REPRODUCTION SETTINGS

In the main experiments, we compared our proposed BAN and PICK strategies against several existing baselines. For brevity, the main text only provided a concise overview of these methods. In this appendix, we present a more detailed description of each compared method, together with the specific reproduction settings used in our experiments. We organize the discussion into two parts: methods compared with PICK, and methods compared with BAN.

### A.7.1 COMPARED METHODS FOR PICK

**Dynamic Routing.** (Huang et al., 2024a) first proposed the idea of dynamically allocating experts, arguing that assigning a fixed number of experts to each token is overly rigid. Instead, the number of experts should be adapted according to the difficulty of the underlying task: tokens from challenging tasks should be allocated more experts, while tokens from easier tasks can be served with fewer. Concretely, experts are sequentially selected in descending order of routing weight until the cumulative weight exceeds a predefined threshold $\tau$. Formally, let $r_{(1)}, r_{(2)}, \ldots, r_{(K)}$ denote the routing weights of experts sorted in descending order. The number of selected experts $k$ is determined as

$$k = \min\Big\{ m \,\Big|\, \sum_{j=1}^{m} r_{(j)} \geq \tau \Big\}.$$

In our experiments, since the optimal threshold $\tau$ cannot be known in advance, we evaluate three candidate values $\tau \in \{0.7, 0.8, 0.9\}$ and report in the main text the result corresponding to the setting that achieves the best average accuracy across benchmarks.

**TiP.** The core idea of TiP is motivated by the observation that, during chain-of-thought (CoT) reasoning, models often switch reasoning paths too frequently in the early exploration stage. Such premature abandonment of a correct path is referred to as *underthinking*. To mitigate this issue, TiP introduces a penalty on tokens that typically indicate a switch of reasoning path (e.g., "alternatively"). Concretely, after the softmax operation, the probability mass of these tokens is reduced by subtracting a fixed penalty value from their logits, thereby encouraging the model to continue deepening its current reasoning path rather than switching too early.

This method involves two hyperparameters: $\alpha$ (penalty strength) and $\beta$ (penalty duration). The original paper did not provide an automatic tuning strategy but instead conducted a grid search to

identify relatively effective values. Following their reported settings, we set $\alpha = 3$ and $\beta = 600$ in our reproduction.

**Rice.** Rice builds on the observation that recent MoE-LLMs often employ an explicit `<think>` token to train chain-of-thought (CoT) reasoning ability. For example, the Qwen3 series can switch between deep-thinking and non-thinking modes by toggling the use of the `<think>` token. Rice leverages this property by comparing expert activation frequencies with and without the `<think>` token, thereby identifying cognitive experts that are closely associated with deep reasoning. Once identified, the routing weight of a cognitive expert is multiplied by a constant factor greater than 1 whenever it is selected, amplifying its influence and enhancing the model's reasoning ability.

Since this method relies on explicit `<think>` tokens, it is only applicable to models such as the Qwen3 series, but not to other models used in our experiments. For reproduction, we followed the original paper's settings: for Qwen3-235B-A22B, we directly used the cognitive experts reported therein; for Qwen3-30B-A3B, as the original paper did not list them, we applied the ICE (Identify Cognitive Experts) procedure proposed in the paper to locate the corresponding cognitive experts before applying the enhancement.

### A.7.2 COMPARED METHODS FOR BAN

**DES.** DES (Dynamic Expert Skip) was proposed to dynamically adjust the number of active experts by exploiting the observation that, for tokens that are relatively easy to process, the routing weights of the selected experts often differ significantly. In such cases, the contribution of the lower-weighted expert to the final output is negligible. Based on this, DES introduces a dynamic pruning strategy: whether to keep the lower-weighted expert is decided according to the ratio between expert weights. For example, in Mixtral-8×7B (Jiang et al., 2024) where each token selects two experts, DES first computes the median ratio of top-1 to top-2 expert weights on a calibration set. During inference, if the actual ratio exceeds this median, the top-2 expert is skipped.

In our reproduction, we aimed to align the pruning degree and acceleration ratio with our proposed Ban method, so that the comparison mainly reflects differences in accuracy preservation. For Qwen3 models, when $\lambda = 0.7$, the average number of selected experts under Ban is slightly below 5. Therefore, we set the maximum number of experts to 6 and the minimum to 4. Specifically, we computed the medians of the ratios $\frac{r_{(4)}}{r_{(5)}}$ and $\frac{r_{(5)}}{r_{(6)}}$ on a calibration set, where $r_{(j)}$ denotes the $j$-th largest routing weight. At inference time, if the first ratio exceeds its median, only the top-4 experts are kept; otherwise, if the first ratio does not exceed its median but the second does, the top-5 experts are kept; otherwise, all top-6 experts are retained. For the DeepSeek models, we followed an analogous procedure to ensure comparable pruning degree and speedup.

**ODP.** ODP (Online Dynamic Pruning) is an extension of DES that incorporates an additional key-token protection mechanism. While DES prunes experts dynamically based on weight ratios, ODP further observes that certain tokens carry disproportionately high importance in the attention map. If a token receives an attention score significantly larger than the others, pruning its experts may harm model performance. To address this, ODP protects such key tokens: even if their expert weights satisfy the pruning condition, they are exempted from pruning.

In our reproduction, we followed exactly the same procedure as DES for the pruning part. On top of that, we adopted the evaluation metric for token importance proposed in the ODP paper to identify key tokens. Whenever a token is classified as key, we enforced the selection of all chosen experts for it, regardless of the pruning condition.

### A.8 CASE STUDY

While the previous experiments have demonstrated the effectiveness of **Pick** in improving model performance across various benchmarks, the results there are largely summarized at an aggregate level. In this section, we take a complementary perspective and conduct two case studies. To make the analysis clear and accessible, we deliberately select examples from the **Math-500** dataset: the problems here are of moderate difficulty, making them easier to understand and to analyze how Pick concretely helps the model succeed. In contrast, instances from high-level competition datasets such as AIME or GPQA-Diamond are often too specialized for non-expert readers (including the authors)

Table 10: Case study 1: An example from Deepseek-v2-Lite-Chat. Red highlights mark incorrect key steps, while green highlights show the corrected reasoning.

| **Problem** | Consider the geometric sequence $\frac{125}{9}, \frac{25}{3}, 5, 3, \ldots$. What is the eighth term of the sequence? Express your answer as a common fraction. 
 Remember to put your final answer within `box`. |
|---|---|
| **DeepSeek-V2-Lite-Chat** | The common ratio of this geometric sequence is $\frac{25/3}{125/9} = \frac{1}{5}$. 
 To find the eighth term, we can use the formula for the $n$th term of a geometric sequence: $a_n = a_1 \cdot r^{(n-1)}$, where $a_n$ is the $n$th term, $a_1$ is the first term, $r$ is the common ratio, and $n$ is the term number. 
 Substituting the given values into the formula, we get: 
 $a_8 = \frac{125}{9} \cdot \left(\frac{1}{5}\right)^{8-1}$ 
 $= \frac{125}{9} \cdot \left(\frac{1}{5}\right)^{7}$ 
 $= \frac{125}{703125}$ 
 $= \boxed{\frac{1}{5625}}$ |
| **+Pick** | The common ratio of this geometric sequence is $\frac{25/3}{125/9} = \frac{3}{5}$. 
 To determine the eighth term, we apply the formula for the $n$th term of a geometric progression: $a_n = a_1 \cdot r^{(n-1)}$, where $a_n$ is the $n$th term, $a_1$ is the first term, $r$ is the common ratio, and $n$ marks the position of the term. 
 Plugging the given values into the expression, we get: 
 $a_8 = \frac{125}{9} \cdot \left(\frac{3}{5}\right)^{8-1}$ 
 $= \frac{125}{9} \cdot \left(\frac{3}{5}\right)^{7}$ 
 $= \frac{125}{9} \cdot \frac{2187}{78125}$ 
 $= \boxed{\frac{243}{625}}$ |

to fully interpret. Instead of reporting overall accuracy, these examples dive into the reasoning process of the models, illustrating concrete instances where the base model initially fails but succeeds after enhancing key experts with Pick. Through these analyses, we aim to provide a more nuanced understanding of how Pick leads to accuracy gains, and to shed light on directions for future work.

**Case1.** This case comes from **Math-500 level 1**, which represents the easiest tier of the benchmark. The task (see Table 10) is to compute the eighth term of a given geometric sequence, a problem that primarily requires identifying the common ratio and then applying the standard formula for the $n$th term. The model DeepSeek-V2-Lite-Chat, which has relatively weaker reasoning capacity, fails here due to a simple computational mistake: it miscalculates the common ratio, and this error propagates through the subsequent steps, even though the overall reasoning structure was essentially correct. With the help of **Pick**, the key experts specializing in mathematical computation are activated. As a result, the model correctly identifies the common ratio and, building upon its otherwise correct reasoning process, arrives at the correct final answer. This exemplifies one of the typical strengths of the Pick method: enhancing a model's ability to perform basic yet crucial calculations.

**Case2.** This case comes from **Math-500 level 3**, which corresponds to middle- to high-school mathematics. The task (see Table 11) was solved using Qwen3-30B-A3B, a model with relatively strong reasoning capabilities. The problem asks for the number of students who take physics, given multiple overlapping counts of students taking calculus, chemistry, and physics. The key challenge lies in correctly interpreting the phrase: "Only 30 take both physics and calculus." In the original response, the model misread the "only" as describing students who *only take physics and calculus but not chemistry*, thereby inferring that the total for the calculus–physics overlap should be $30 + 15 = 45$. This misunderstanding propagated through the inclusion–exclusion calculation and led to the wrong final answer. With the help of **Pick**, however, the model activated experts capable of parsing such subtle linguistic cues, and correctly understood that "only" here contrasts with the other number (75 in physics–chemistry), so the total intersection of calculus and physics is just 30 (already including

Table 11: Case study 2: An example from Qwen3-30B-A3B. Because the model's output is very long, we only retain the key reasoning steps and mark omitted content by ellipses (...). Red highlights mark incorrect key steps, while green highlights show the corrected reasoning.

| Problem | There are 360 people in my school. 15 take calculus, physics, and chemistry, and 15 don't take any of them. 180 take calculus. Twice as many students take chemistry as take physics. 75 take both calculus and chemistry, and 75 take both physics and chemistry. Only 30 take both physics and calculus. How many students take physics? Remember to put your final answer within box. |
|---|---|
| Qwen3-30B-A3B | Okay, let's try to figure out how many students take physics in this school. There are 360 in total, 15 take none, so 345 take at least one subject. ... |
| | Recall inclusion-exclusion: $$|C \cup P \cup Ch| = |C| + |P| + |Ch| - |C \cap P| - |C \cap Ch| - |P \cap Ch| + |C \cap P \cap Ch|$$ |
| | We know $|C| = 180, |C \cap Ch| = 75, |P \cap Ch| = 75, |C \cap P \cap Ch| = 15, Ch = 2P$. ... |
| | Now about "Only 30 take both physics and calculus." **The phrase suggests this is exactly physics+calculus (not including chemistry). Therefore total $|C \cap P| = 30 + 15 = 45$.** |
| | So plug into formula: $$345 = 180 + P + 2P - 45 - 75 - 75 + 15$$ |
| | Simplify constants $\ldots \Rightarrow 3P = 345 \Rightarrow P = 115$ |
| | **Final Answer:** $\boxed{115}$ |
| +Pick | Okay, let's try to figure out how many students take physics in this school. Total students 360, minus 15 none = 345 take at least one. ... |
| | Again apply inclusion-exclusion: $$|C \cup P \cup Ch| = |C| + |P| + |Ch| - |C \cap P| - |C \cap Ch| - |P \cap Ch| + |C \cap P \cap Ch|$$ |
| | Known: $C = 180, C \cap Ch = 75, P \cap Ch = 75, C \cap P \cap Ch = 15, Ch = 2P$. ... |
| | Now for "Only 30 take both physics and calculus." **So $|C \cap P| = 30$ (this already includes the 15 in all three).** |
| | Substitute: $$345 = 180 + P + 2P - 30 - 75 - 75 + 15$$ |
| | Simplify constants $\ldots \Rightarrow 3P = 330 \Rightarrow P = 110$ |
| | **Final Answer:** $\boxed{110}$ |

those in all three). As a result, the reasoning proceeded correctly and the right answer was obtained. This case illustrates a different aspect of the Pick method: besides arithmetic accuracy, it can also enhance a model's general interpretive and linguistic reasoning ability, allowing it to disambiguate tricky instructions in math word problems.

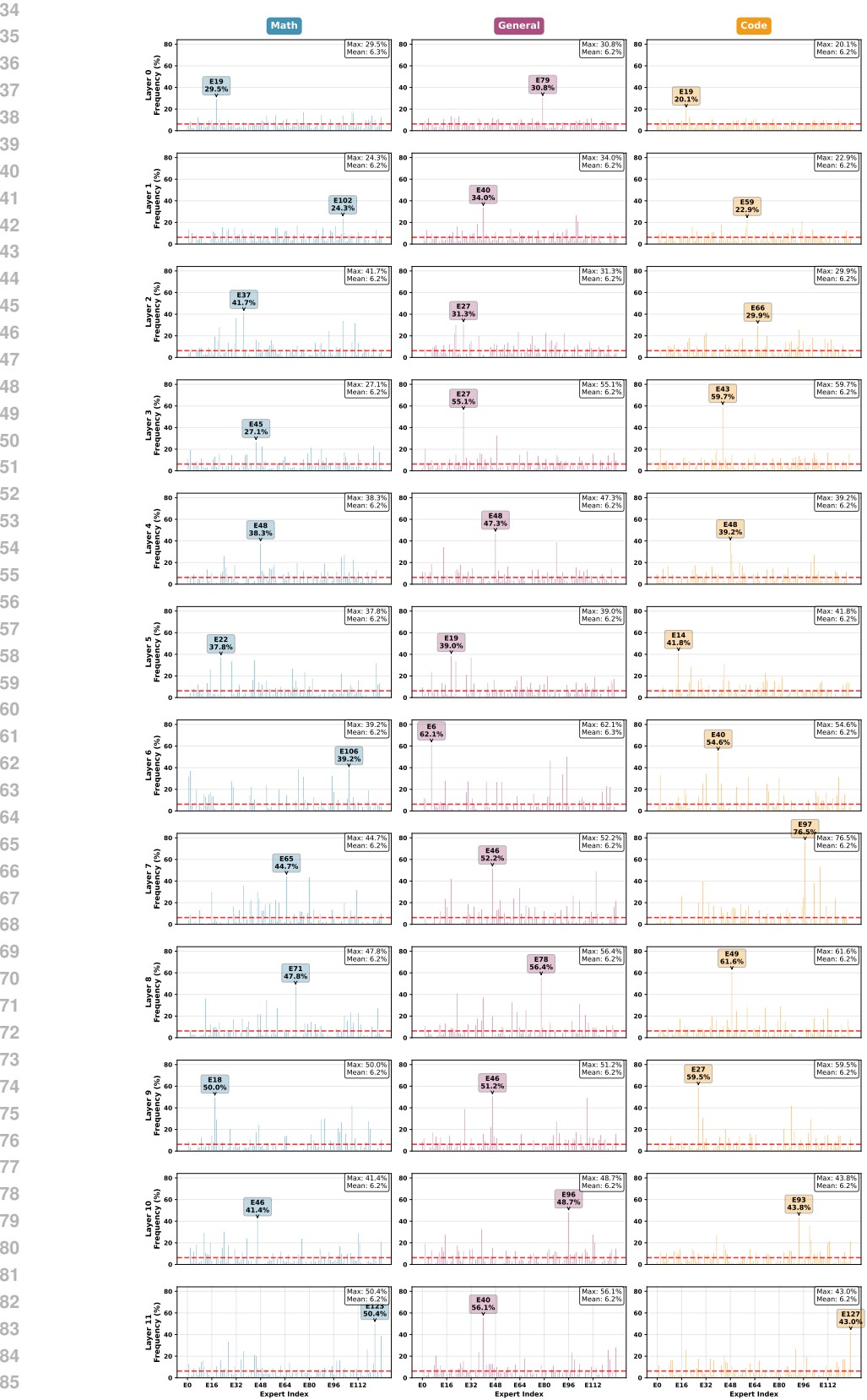

Figure 6: Expert usage frequency for layers 0–11 across math, code, and general tasks.

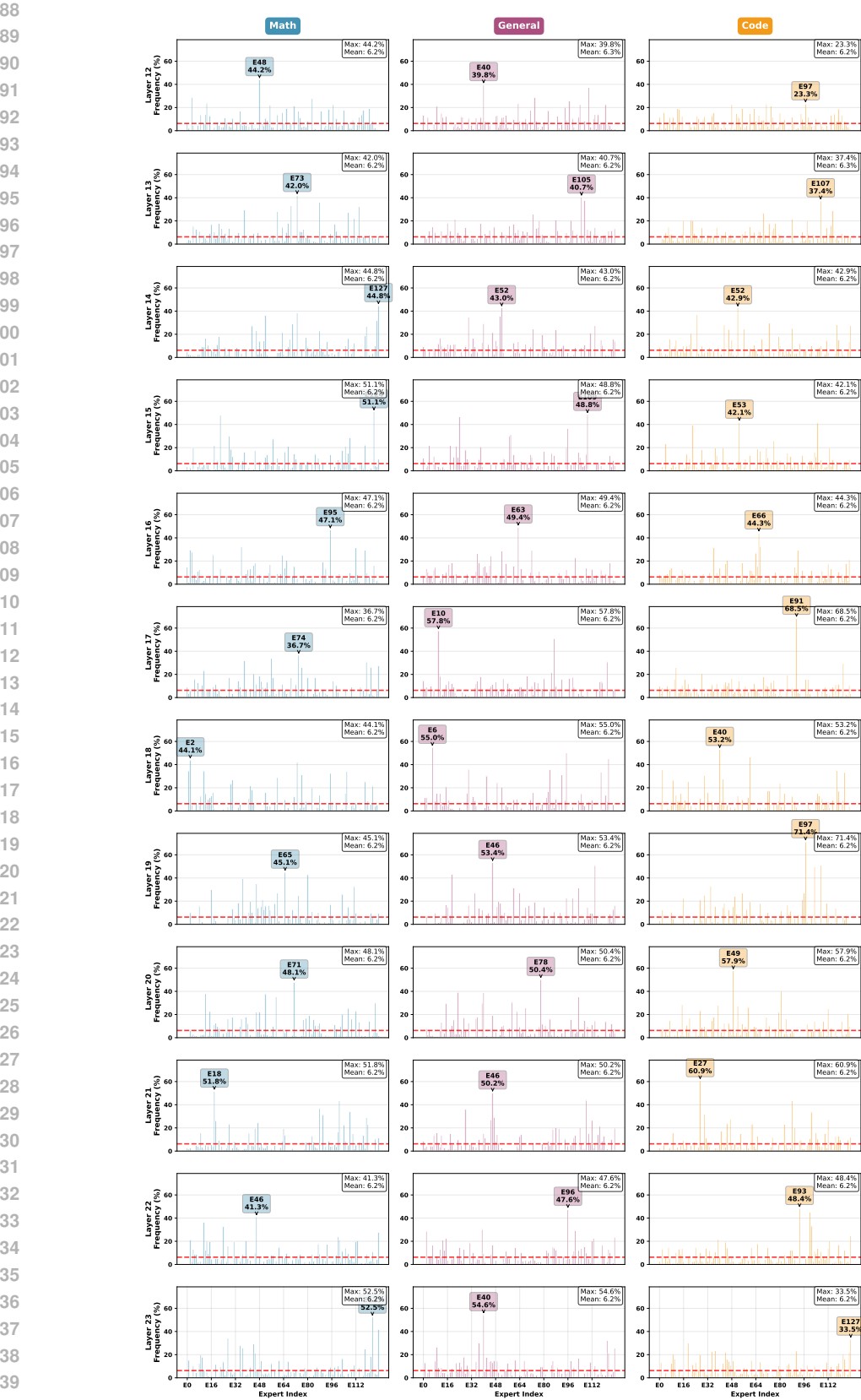

Figure 7: Expert usage frequency for layers 12–23 across math, code, and general tasks.

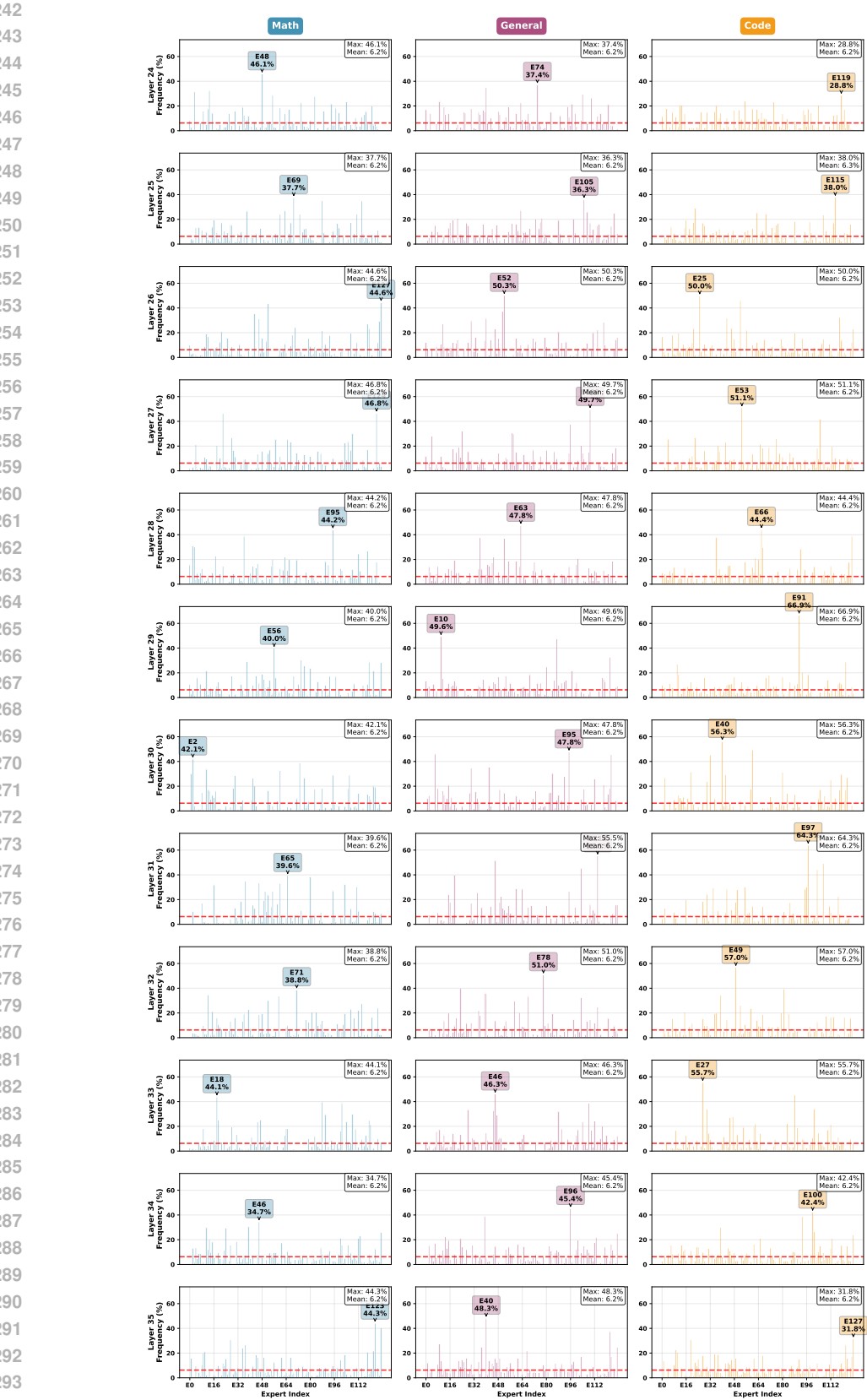

Figure 8: Expert usage frequency for layers 24–35 across math, code, and general tasks.

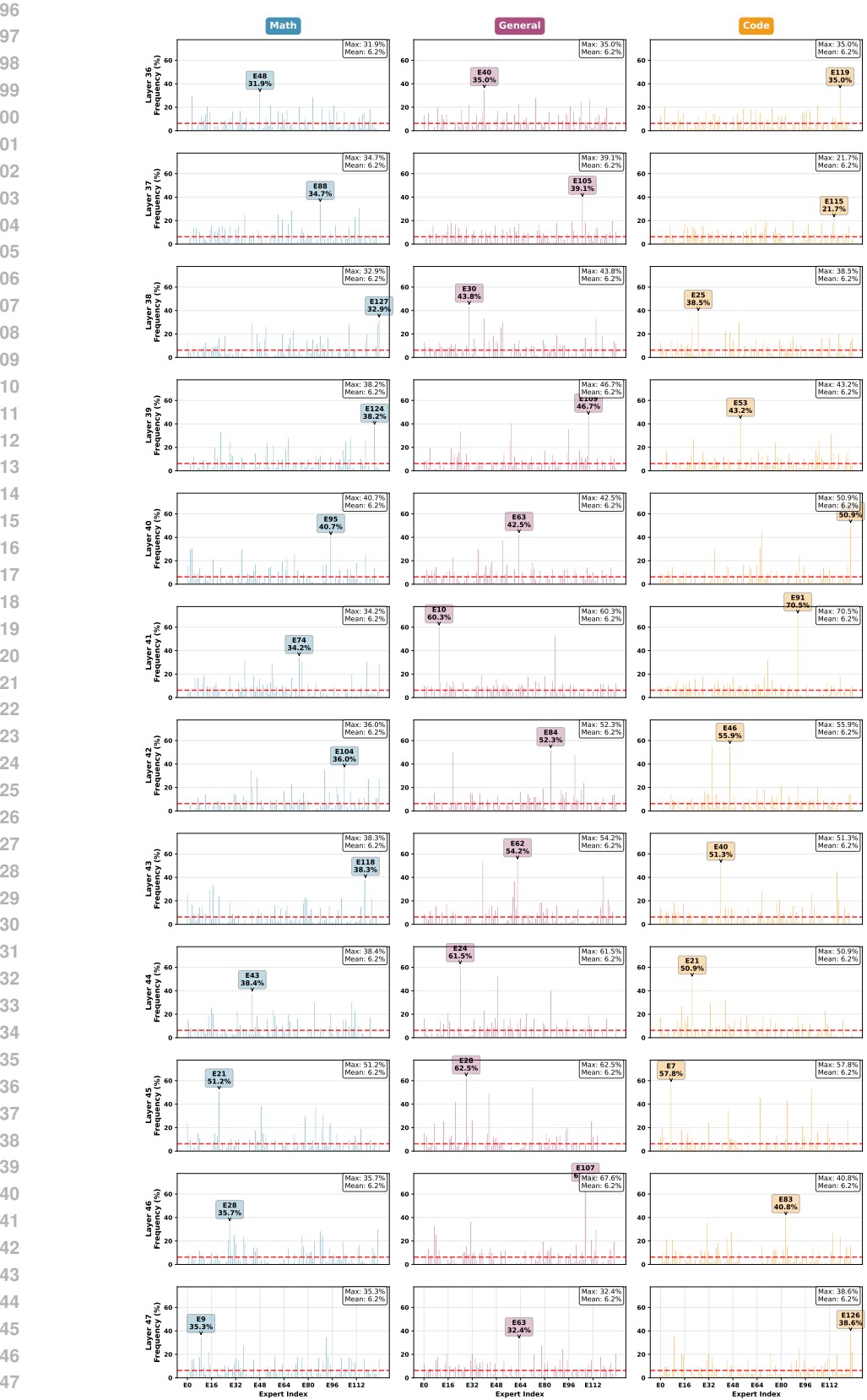

Figure 9: Expert usage frequency for layers 36–47 across math, code, and general tasks.

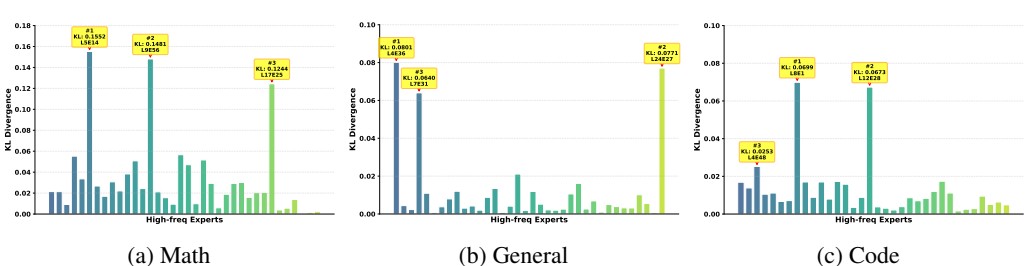

Figure 10: KL divergence of domain-specific experts when pruned, for **DeepSeek-V2-Lite-Chat** across math, general, and code tasks.

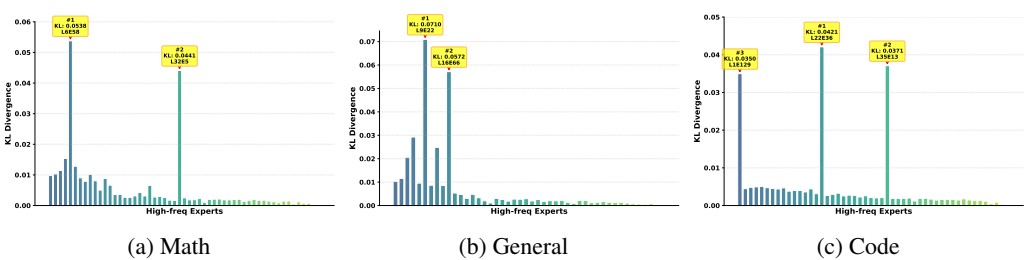

Figure 11: KL divergence of domain-specific experts when pruned, for **DeepSeek-v2.5-1210** across math, general, and code tasks.

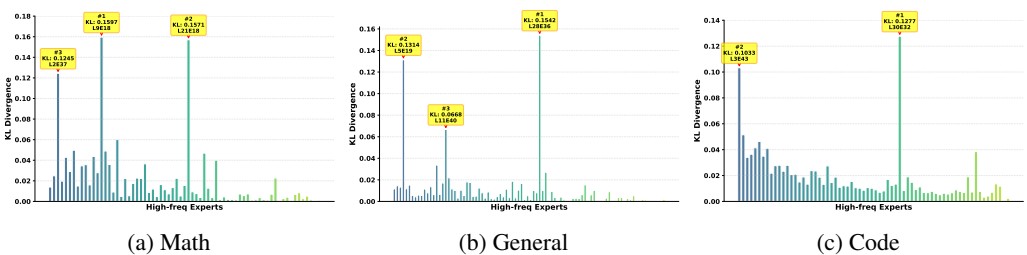

Figure 12: KL divergence of domain-specific experts when pruned, for **Qwen3-30B-A3B** across math, general, and code tasks.

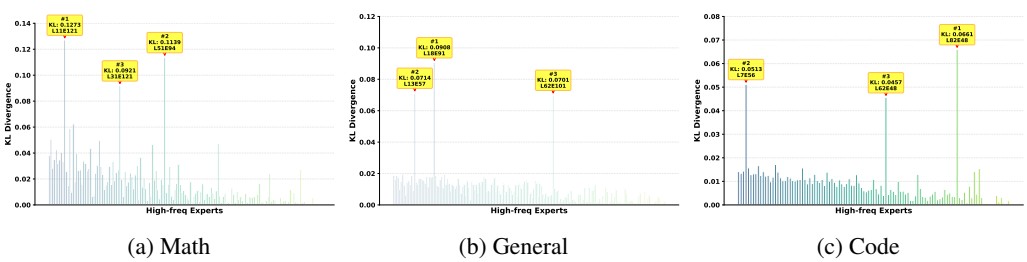

Figure 13: KL divergence of domain-specific experts when pruned, for **Qwen3-235B-A22B** across math, general, and code tasks.

---

**Algorithm 1** Ban: Dynamic Expert Pruning for Fine-grained MoE

---

1: **Inputs:**
    Fine-grained MoE model with $L$ layers
    Base number of experts per token $K_{\text{base}}$
    Minimum number of experts $K_{\text{min}}$
    Layer sensitivity scores $\{W_l\}_{l=1}^L$
    Token statistics $(R_{\text{min}}, R_{\text{max}})$ from calibration
    Pruning coefficient $\lambda \in (0, 1)$
2: **Offline: normalize layer sensitivity**
3: $W_{\text{min}} \leftarrow \min_l W_l, \quad W_{\text{max}} \leftarrow \max_l W_l$
4: **for** $l = 1$ to $L$ **do**
5:     $L_l' \leftarrow (W_l - W_{\text{min}})/(W_{\text{max}} - W_{\text{min}})$                 $\triangleright\ L_l' \in [0, 1]$
6: **end for**
7: **Online: dynamic pruning at inference**
8: **for** each input sequence **do**
9:     **for** $l = 1$ to $L$ **do**
10:         Perform MoE routing at layer $l$ to obtain expert weights $w_{i,l,e}$ for each token $i$
11:         **for** each token $i$ **do**
12:             Sort experts by $w_{i,l,e}$ in descending order and obtain $e^{(1)}, \ldots, e^{(K_{\text{base}})}$
13:             $S_{i,l}^{(K_{\text{base}})} \leftarrow \sum_{k=1}^{K_{\text{base}}} w_{i,l,e^{(k)}}$
14:             $S_{i,l}^{(K_{\text{min}})} \leftarrow \sum_{k=1}^{K_{\text{base}}} w_{i,l,e^{(k)}} \cdot \mathbb{I}[k \leq 3]$
15:             $R_{i,l} \leftarrow S_{i,l}^{(K_{\text{min}})}/S_{i,l}^{(K_{\text{base}})}$
16:             $T_{i,l}' \leftarrow (R_{\text{max}} - R_{i,l})/(R_{\text{max}} - R_{\text{min}})$     $\triangleright\ T_{i,l}' \in [0, 1]$
17:             $S_{i,l} \leftarrow \lambda \cdot (L_l' + T_{i,l}')/2$
18:             $K_{i,l} \leftarrow \text{round}\big(K_{\text{min}} + (K_{\text{base}} - K_{\text{min}}) \cdot S_{i,l}\big)$
19:             Keep only the top-$K_{i,l}$ experts for token $i$ at layer $l$ according to $w_{i,l,e}$
20:         **end for**
21:         Compute the MoE layer output using the pruned expert sets
22:     **end for**
23: **end for**

---

## A.9   PSEUDOCODE FOR BAN

In this subsection, we present the pseudocode of the proposed BAN algorithm for dynamic expert pruning in fine-grained MoE models. Our aim is to make the procedure explicit and reproducible, and to clearly illustrate how layer-wise and token-wise sensitivities are combined to determine the number of active experts for each token at each layer. For clarity, the pseudocode is written with an explicit per-token loop; in practice, all token-wise operations are implemented using efficient vectorized (batched) kernels rather than literal for-loops.

## A.10   A.10 DISCUSSION ON STRATEGY SELECTION FOR PICK ENHANCEMENT

In the main text (Section 4.3), we showed through experiments on MATH-500, AIME2024, and AIME2025 that among the five proposed enhancement strategies for key experts, **Strategy D** achieves the most consistent accuracy improvements. Here, we provide a more detailed explanation of the motivations behind the design of these strategies and the reasoning that led us to ultimately adopt Strategy D.

**Initial motivation: Strategies A and B.**   An initial intuition was straightforward—since key experts play a decisive role in task-specific reasoning, one might expect model performance to improve if all tokens are explicitly routed through or enhanced by these key experts. Based on this idea, **Strategies A** and **B** were designed to directly enforce the participation of key experts for all tokens, aiming to strengthen their influence across the entire sequence regardless of token type or context. This approach represents the most direct attempt to leverage key experts' strong impact on output generation to achieve better overall results.

**Why Strategies A and B are not stable.**   However, subsequent analyses in Sections 4.1 and 4.2 of the main paper suggest that key experts are closely tied to task-relevant tokens and can be regarded as a subset of domain-specialized experts. Taking the *math* domain as an example, these key experts are highly activated by digits, letters, and operators, but not by unrelated tokens such as pronouns or prepositions. Consequently, forcing key experts to be enhanced for every token disregards this natural specialization and leads to unstable or degraded performance across datasets. This behavior is **consistent with the experimental results in Section 4.3**, where Strategies A and B exhibit larger performance fluctuations and reduced accuracy compared with other strategies. These observations confirm that a more adaptive mechanism is needed to selectively enhance key experts where they are most beneficial, rather than applying uniform enforcement.

**Adaptive Enhancement Yields Better Performance.**   To address the instability observed in Strategies A and B, we designed adaptive strategies that selectively enhance key experts based on token-level routing signals rather than enforcing fixed participation. A natural idea might be to construct an explicit lookup table of task-relevant tokens so that enhancement can be applied in a perfectly fine-grained manner. However, such an approach would be overly cumbersome in implementation and inconsistent with our design objective of maintaining simplicity and efficiency. Instead, we empirically observed that for tokens within a key expert's corresponding domain—for example, math-related tokens—the key experts often appear within the top-$2k$ routing weight range even when they are not selected in the top-$k$ ($k$ is the default number of experts per token), whereas for unrelated tokens they seldom appear within the top-$2k$. This observation provides a lightweight and effective criterion for identifying which tokens should trigger the enhancement of their domain-specific experts.

We conducted this verification on QWEN3-30B-A3B using the math key expert $(2, 37)$. All tests were performed on the *math calibration subset*. To analyze cross-domain behavior, we selected the top-20 most frequent tokens from both the *math* and *general* word clouds shown in Figure 2 of the main paper. For each token, we measured the frequency with which the math key expert appears in the top-$k$, in the top-$2k$ but not in the top-$k$, and outside the top-$2k$ routing weight range. The results are presented in Table 12.

Table 12: Routing frequency (%) of the math key expert $(2, 37)$ on the math calibration set. Tokens are taken from the *math* and *general* word clouds in Figure 2. Each column shows the proportion of tokens whose key expert appears in different routing weight ranges.

| Token Type (20 tokens each) | Top-$k$ (%) | Top-$2k$ but not Top-$k$ (%) | Outside Top-$2k$ (%) |
|---|---|---|---|
| Math tokens | 68.4 | 24.7 | 6.9 |
| General tokens | 2.5 | 0.4 | 97.1 |

The results show that, even when evaluated purely on the math calibration set, the math key expert $(2, 37)$ is highly activated for math-related tokens—appearing in the top-$2k$ routing range for more than 90% of them—while rarely activated for general tokens. This indicates that the routing mechanism naturally captures a strong correlation between math tokens and the math key expert, validating

that the adaptive strategy can precisely align experts with domain-relevant tokens. Consequently, **Strategies C** and **D** leverage this intrinsic property to enhance key experts selectively, improving performance on relevant tokens without blind strengthening across the vocabulary.

**Choice between Strategy C and D.** The primary difference between Strategies C and D lies in whether the minimally weighted selected expert is replaced. Our observations indicate that the least-weighted expert contributes negligibly to model performance, consistent with the motivation of the **Ban** module: pruning or replacing experts of minimal contribution improves efficiency with minimal accuracy loss. Thus, replacing or retaining this expert makes little difference; we therefore adopted **Strategy D** for its simplicity and implementation efficiency.

**Future directions.** We believe this analysis offers deeper insight into the design rationale of Pick. Future extensions may incorporate token-adaptive or data-driven policies to automatically determine enhancement conditions, potentially providing finer control and better synergy with the underlying Ban mechanism while maintaining the plug-and-play nature of the framework.

## A.11 SYSTEM-LEVEL PERFORMANCE EVALUATION

To conduct a more detailed system-level performance evaluation of our proposed **Ban&Pick** method, we performed additional experiments on the **Qwen3-30B-A3B** model. The evaluation focuses on **throughput** and **memory usage**, which are important metrics for assessing the efficiency of large-scale models in real deployment scenarios.

All tests were executed on a single A800-80G GPU. We used identical prompts and identical inference configurations for both the base model and the Ban&Pick-enhanced model (*maximum output length* = 1k, two warm-up runs, and five averaged runs). The average throughput and peak memory usage are summarized in Table 13.

Table 13: System-level evaluation of Ban&Pick on Qwen3-30B-A3B.

| Metrics | Base | +Ban&Pick | Change |
|---|---|---|---|
| Avg Top-$k$ | 8.00 | 4.82 | $-39.75\%$ |
| Throughput (tokens/s) | 10.01 | 12.85 | $+28.37\%$ |
| Peak Memory Usage (GB) | 5.84 | 5.32 | $-8.90\%$ |

The results show that, due to expert pruning introduced by the Ban module, **throughput increases notably by +28.37%**, while **peak memory usage decreases slightly**. These results demonstrate that Ban&Pick not only improves model accuracy but also enhances **efficiency and deployability** in practical inference scenarios.

