# OpenReview forum: "Ban&Pick: Ehancing Performance and Efficiency of MoE-LLMs via Smarter Routing"
_ICLR.cc/2026/Conference — Submitted to ICLR 2026_

### Official Review · Reviewer_364k · 2025-10-29

**Soundness:** 3
**Presentation:** 3
**Contribution:** 3
**Rating:** 8
**Confidence:** 3

**Summary:**

This paper proposes Ban&Pick, a post-training, plug-and-play framework that improves both the performance and efficiency of fine-grained Mixture-of-Experts (MoE) large language models. The method introduces two complementary modules: Pick, which identifies and reinforces key experts with major impact on task performance, and Ban, which dynamically prunes redundant experts based on layer and token sensitivity.

Without retraining or architectural modification, the framework achieves consistent accuracy improvements and inference acceleration across multiple MoE-LLMs (DeepSeek, Qwen3) on math, code, and reasoning benchmarks. The work is technically sound and demonstrates strong engineering novelty and practical value.

**Strengths:**

1. Demonstrates a clever engineering insight—leveraging untapped routing inefficiencies to boost performance and efficiency simultaneously.

2. Provides a lightweight, generalizable, and plug-and-play approach applicable to existing fine-grained MoE architectures.

3. The empirical validation is comprehensive, spanning multiple large models and diverse domains.

4. Shows clear, measurable engineering gains (up to +3% accuracy and 1.25× speedup) without any additional training cost.

5. Strong potential for real-world adoption in industrial MoE inference systems (e.g., vLLM, DeepSeek).

**Weaknesses:**

1. Limited theoretical depth. The paper focuses on empirical effectiveness but does not deeply analyze the underlying mechanism of key expert influence or convergence behavior.

2. Lack of comparison with more recent MoE optimization baselines, such as MoE++, adaptive gating, or mixed routing approaches (e.g., Mixtral, Soft-MoE).

3. Evaluation scope is somewhat narrow, emphasizing accuracy and speedup but lacking broader system-level metrics like throughput, latency variance, or memory usage.

4. Minor stylistic and formatting issues, including small typos and missing pseudocode for algorithmic clarity.

5. Noticeable typographical issue: the paper title itself contains a serious typo (“EHANCING” should be “ENHANCING”), which reflects carelessness in proofreading.

**Questions:**

1. Can the proposed Ban&Pick framework be combined with adaptive routing methods or reinforcement-learning-based expert selection?

2. How sensitive is the method to hyperparameters (λ, Kmin) across different model scales?

3. Could online or task-specific adaptation further enhance the framework’s performance?

---

> ### Author Response · Authors · 2025-11-17
> **Part 1/3 (W1~2)**
>
> Dear reviewer 364k,
>
> We thank the reviewer for the positive overall assessment of our work, especially recognizing its practical value, engineering novelty, and potential for real-world deployment. We also appreciate the thoughtful suggestions on theory, baselines, and system-level evaluation, and we are grateful for the reviewer’s tolerance of minor typos in the current version. Below we will address the raised questions and concerns one by one.
>
> > W1: Limited theoretical depth. The paper focuses on empirical effectiveness but does not deeply analyze the underlying mechanism of key expert influence or convergence behavior.
>
> Regarding the theoretical depth, we have substantially expanded the analysis in the revised version. In particular, **Section 8 (pp. 9–10, highlighted in blue)** now provides a more systematic explanation of (i) why fine-grained MoE tends to under-utilize key experts and (ii) why enhancing these key experts is effective. Building on the interaction between premature routing and the auxiliary balancing loss, we analyze how this combination constrains expert specialization during pretraining and leads to key experts whose actual importance is not fully reflected in their routing probabilities, thereby motivating our Pick design.
>
> We believe that, based on these analyses, future work can further explore how to better leverage key experts by jointly studying their roles during both pretraining and inference.
>
>
>
> > W2: Lack of comparison with more recent MoE optimization baselines, such as MoE++, adaptive gating, or mixed routing approaches (e.g., Mixtral, Soft-MoE).
>
> Thank you for pointing out these recent MoE optimization methods. We have carefully read works such as MoE++, adaptive gating, and mixed-routing approaches (e.g., Mixtral, Soft-MoE). While they share with us the overarching goal of improving expert routing, they mainly act at **pretraining time** by modifying the architecture and/or training objectives. In contrast, our method is explicitly designed as an **inference-time**, plug-and-play framework that does not alter the MoE architecture or its pretrained weights.
>
> Because of this difference in optimization stage and setting, a direct head-to-head comparison may not be entirely apples-to-apples. At the same time, we view these approaches as largely **complementary** to ours rather than competing: in principle, models trained with MoE++, adaptive gating, or mixed routing could further benefit from applying Ban&Pick at inference time.

---

> ### Author Response · Authors · 2025-11-17
> **Part 2/3 (W3~5)**
>
> > W3: Evaluation scope is somewhat narrow, emphasizing accuracy and speedup but lacking broader system-level metrics like throughput, latency variance, or memory usage.
>
> Thank you for this valuable suggestion. Following your recommendation, we conducted additional experiments on **Qwen3‑30B‑A3B** to evaluate **throughput** and **memory usage**. The tests were performed on a single **A800‑80G** GPU. Using identical prompts, we compared the original model and the Ban&Pick‑enhanced model under the same configuration (maximum output length = 1k, two warm‑up runs, and five averaged runs). We recorded the average throughput and peak memory usage, as summarized in the table below.
>
> |        Metrics         | Base  | +Ban&Pick | Change  |
> | :--------------------: | ----- | --------- | ------- |
> |       Avg Top‑k        | 8     | 4.83      | −39.75% |
> | Throughput (tokens/s)  | 10.01 | 12.85     | +28.37% |
> | Peak Memory Usage (GB) | 58.4  | 53.2      | −8.90%  |
>
> The results show that, due to expert pruning in the Ban module, throughput increases notably by **+28.37%**, and peak memory usage is slightly reduced, making the model more deployment‑friendly and efficient in practical inference scenarios.
>
>
>
> > W4 and W5: Minor stylistic and formatting issues, including small typos and missing pseudocode for algorithmic clarity. Noticeable typographical issue: the paper title itself contains a serious typo (“EHANCING” should be “ENHANCING”), which reflects carelessness in proofreading.
>
> We sincerely thank the reviewer for carefully pointing out the typographical, stylistic, and formatting issues. We have carefully reviewed the entire paper and corrected spelling errors and formatting inconsistencies. Following your suggestion, we also added a new section (**Appendix A.9**) in the revised version to provide pseudocode for the Ban algorithm for better algorithmic clarity; all corresponding changes are highlighted in **green** in the updated PDF.
>
> We also sincerely apologize for the typographical error in the title and will request a title correction in the submission system. We greatly appreciate the reviewer’s understanding and attention to these details.

---

> ### Author Response · Authors · 2025-11-17
> **Part 3/3 (Q1~3)**
>
> > Q1: Can the proposed Ban&Pick framework be combined with adaptive routing methods or reinforcement-learning-based expert selection?
>
> Yes, the proposed Ban&Pick framework can be combined with adaptive routing methods, and potentially also with reinforcement‑learning‑based expert selection.
>
> If the adaptive routing (or RL‑based expert selection) method is used at the **pretraining stage**, integration is straightforward: Ban&Pick operates purely at **inference time** as a plug‑and‑play module and does not modify the pretrained architecture, routing network, or expert weights. Models trained with such methods could directly serve as the backbone on which Ban&Pick is applied.
>
> If adaptive routing is applied **during inference**, it can also be compatible in many cases. Such methods can often be viewed as a form of dynamic expert pruning or reweighting, and can be combined with Ban&Pick by incorporating the **sensitivity analysis from Ban**—in particular, layer sensitivity and token sensitivity. For example, under a fixed cumulative routing‑weight threshold, one could assign **higher thresholds to more sensitive layers/tokens**, effectively merging adaptive routing behavior with Ban’s structured control. In this way, Ban&Pick can act as a systematic controller on top of existing adaptive routing or RL‑based selection strategies, rather than replacing them.
>
>
>
> > Q2: How sensitive is the method to hyperparameters (λ, Kmin) across different model scales?
>
> As discussed in **Appendix A.5**, we use **Qwen3‑30B‑A3B** as an example to illustrate both the straightforward selection of $K_{\min}$ and the robustness of the pruning‑control hyperparameter $\lambda$. The results show that $K_{\min}$ effectively enforces a lower bound on the number of selected experts, and that the Ban module remains robust to variations in $\lambda$ in terms of both accuracy and speedup.
>
> To further verify the robustness of these hyperparameters across **different model scales**, we additionally conducted experiments on **Qwen3‑235B‑A22B**. We varied $\lambda$ in {0.7, 0.8, 0.9} and evaluated the average number of selected experts, accuracy across three domain datasets, and inference speedup. The results (see table below) demonstrate that both accuracy and speedup remain stable across settings.
>
> | λ (in Ban) | Avg Topk | AIME2024 | GPQA  | HumanEval+ | Speedup |
> | :---------: | :------: | -------- | ----- | ---------- | ------- |
> |  (no Ban)   |   8.00   | 84.67    | 71.21 | 78.05      | 1.00    |
> |     0.7     |   4.77   | 83.33    | 69.70 | 75.61      | 1.27    |
> |     0.8     |   5.03   | 83.33    | 70.71 | 76.22      | 1.25    |
> |     0.9     |   5.21   | 83.33    | 70.20 | 76.83      | 1.24    |
>
> These findings suggest that the method is **robust to $\lambda$ across model scales**, and that in practical use, this hyperparameter can be chosen easily within a reasonable range without extensive tuning, while $K_{\min}$ provides a simple and effective safeguard on the minimum level of sparsity.
>
>
>
> > Q3: Could online or task-specific adaptation further enhance the framework’s performance?
>
> This is a very interesting direction—one that we also explored during our experiments. Our findings suggest that, while our main design emphasizes **generality**, **task-specific adaptation** can indeed further boost performance on individual tasks when cross-task generalization is not the primary concern.
>
> For example, in the math domain, the implementation of Pick in our paper adopts a simple, unified enhancement strategy (Strategy D) across all math datasets, enhancing all three identified math-domain key experts (as shown in Table 4). This straightforward approach already improves Qwen3‑30B‑A3B’s accuracy on Math‑500 from 95.2% to **96.6%** (see Figure 4). However, when we selectively enhanced only two key experts—(9, 18) and (21, 18)—the accuracy on Math‑500 further increased to **97.2%**, albeit with slightly smaller gains on AIME2024 and AIME2025.
>
> This observation suggests that the Pick module can be **further adapted in a task-specific or online manner**. We believe that, building upon our framework, future work may explore automatic strategies to dynamically **select or reweight key experts** according to task characteristics, achieving more tailored yet still efficient adaptation when desired.

---

> > ### Comment · Reviewer_364k · 2025-11-25
> >
> > I acknowledge I have read the rebuttal posted by the authors to my review. Thank the authors for the response to my previous comments. I appreciate the time and effort the authors have put into addressing the concerns raised. While the authors provided responses to the raised concerns, given that the necessary revisions fall outside the scope of what can be accommodated between the current version and the camera-ready version, my review and score remain unchanged.

---

> > > ### Author Response · Authors · 2025-11-25
> > >
> > > Thank you for your positive assessment of our proposed Ban&Pick framework. Your insightful suggestions have been very helpful in improving the quality of the paper. We sincerely appreciate the time and effort you have dedicated to our work.

---

### Official Review · Reviewer_Rchp · 2025-10-30

**Soundness:** 3
**Presentation:** 3
**Contribution:** 2
**Rating:** 4
**Confidence:** 4

**Summary:**

The paper proposes Ban&Pick, which is a post-training, plug-and-play strategy for smarter routing for selecting expert in the modern MoE models. The Pick discovers and reinforces key experts that is a small group with outsized impact on performance, which leading to notable accuracy gains across domains. Ban further dynamically prunes redundant experts based on layer and token sensitivity, delivering faster inference with minimal accuracy loss. The author of the paper performs experiments on fine-grained MoELLMs (DeepSeek, Qwen3) across math, code, and general reasoning benchmarks demonstrate that Ban&Pick delivers free performance gains and inference acceleration without retraining or architectural changes.

**Strengths:**

1. The findings on domain-specialized experts and the redundant expert activation is critical to the design of the Ban&Pick. The overall design of the method is very straightforward and easy to implement.

2. The analysis of key expert is very thorough. The explanation is easy to understand. Same writing style is appreciated for redundant expert selection. The whole structure is clear and easy to follow.

3. I believe the discovery is novel and inspiring. Although the method seems to be straightforward, the discovery can be utilized in many future research.

**Weaknesses:**

1. It is not clear to me what is the key reason of failing to include of some key expert in specific domain tasks. According to the earlier discussion in the paper, the reason might come from the premature of the routing strategy during pretraining. Some of the related works also proves that. But in this paper, the author does not show the direct connection between this reason (premature) to the issue (expert selecting failure). The experiments in figure 1, 2, 3, is showing the fact that there are key experts and they are important, but the whole paper fails at showing why. The author of the paper should articulate on this and it will make the paper better by a lot.

2. In order to enhancing the key experts, the author of the paper proposes 5 strategy and the author of the paper uses strategy D. But the reader would be more convincing of why the author of the paper choose such design.

3. Comparing to Pick, the Ban sometimes degrade the accuracy, which is understandable. But combine with Pick, the goal of this paper would become improve the performance of the MoE model. However, the Ban&Pick results in Table 3 shows varying improve. Although the difficulty of the tasks are different, some of the tasks like Math500 has limited improvement. And within the same task like GPQA, different models adapt to the Ban&Pick differently, which shows unstable performance of the method.

**Questions:**

Please refer to weaknesses.

---

> ### Author Response · Authors · 2025-11-17
>
> Dear reviewer Rchp,
>
> Thank you for your thoughtful review and for highlighting the clear and easy-to-implement design of Ban&Pick, the thorough analysis, and the novelty and potential impact of our findings. We are very glad to have the opportunity to discuss, one by one, the three insightful questions you raise in your review.

---

> ### Author Response · Authors · 2025-11-17
> **Part 1/3 (W1)**
>
> > W1: It is not clear to me what is the key reason of failing to include of some key expert in specific domain tasks......
>
> We fully agree that clarifying why some key experts are under-utilized is important. In the revised version, we have added a dedicated discussion in Section 8 (pp. 9–10, highlighted in blue) to address this issue, and we also summarize the main points here:
>
> Our central claim is that both **premature routing** and the **auxiliary balancing loss** jointly lead to insufficient emphasis on key experts in fine-grained MoE. Below, we first review the balancing loss and its behavior in fine-grained MoE, and then explain the resulting under-utilization of key experts based on these.
>
> 1. **Background: auxiliary balancing loss and its effect**
>    In early MoE architectures, each layer typically contained only a small number of experts. Without additional constraints, training tended to collapse onto a few experts that were activated very frequently, while the remaining experts received too few updates and degraded. To address this, prior works [1] introduced an auxiliary balancing loss that explicitly regularizes the router toward more even expert usage. A simplified form of the loss is (eq1):
>
>     $\mathcal{L}_{\text{expert}} = \alpha \sum_i f_i \cdot p_i $
>
>    where $f_i$ is the actual activation frequency of expert i, $p_i$ is the average routing score assigned to expert i, and $\alpha$ is a balancing coefficient. Intuitively, when an expert receives both high routing probability and high actual usage, the contribution $f_i \cdot p_i$ becomes large and is penalized. This mechanism is effective at preventing collapse and widely used in almost all MoE training.
>
>    However, as also noted in follow-up work such as [2], the auxiliary balancing loss conflicts with one of the core goals of MoE: expert specialization. Ideally, the MoE model should be encouraged to develop clearly specialized experts that are responsible for different types of tasks, but the balancing loss discourages this behavior. **As a result, there is an inherent conflict between expert specialization and training stability that is widely present in MoE training.**
>
> 2. **Fine-grained MoE: partial relief but persistent constraints**
>    Fine-grained MoE designs increase the number of experts per layer dramatically. As a result, the summation in Eq (1) spans many more experts, so the influence of each individual expert on the overall term becomes smaller, which allows some experts to have noticeably higher activation frequencies while keeping training relatively stable, as long as these frequencies are not excessively large. Leveraging this property, fine-grained MoE maintains stable training while still enabling a non-trivial degree of expert specialization to emerge.
>
>    However, at the same time, even for strongly domain-specialized experts, their activation frequencies rarely exceed ~50% for a given domain, **indicating that the auxiliary balancing loss still caps how dominant any single expert can become.** These effects are consistent with our empirical analysis in Section 4.1.
>
> 3. **Theoretical explanation: under-utilization of key experts**
>    We now explain how premature routing and auxiliary balancing loss jointly lead to under-utilization of key experts. Conceptually, this process can be divided into two stages during pre-training.
>
>    (1) In the early stage of fine-grained MoE pre-training, before routing prematurely converges, the router already learns a relatively stable mapping from task types to experts: different domains are routed to different subsets of experts, and expert specialization starts to emerge. **However, the auxiliary balancing loss limits how high the activation frequency of any single expert can become for a given domain, preventing overly concentrated routing.**
>
>    (2) When routing becomes premature, the routing-related loss has essentially converged and the routing pattern becomes stable: the selection probabilities for each expert no longer adapt significantly. For the remainder of pre-training, the frequently selected experts for each domain continue to receive large amounts of domain-specific data, so their parameters are updated heavily and they become increasingly specialized for that domain. **As a result, some of these experts evolve into key experts that are particularly important for certain tasks, but their activation frequencies remain capped by the earlier, balance-constrained routing decisions instead of increasing in proportion to their improved competence.** This can also explain why our Pick method works: it compensates for this gap at inference time, by manually enhancing the usage of these key experts so that their activation frequency better matches their actual specialization.
>
> **References:**
> [1] Switch Transformers: Scaling to Trillion Parameter Models with Simple and Efficient Sparsity
> [2] Auxiliary-Loss-Free Load Balancing Strategy for Mixture-of-Experts

---

> ### Author Response · Authors · 2025-11-17
> **Part 2/3 (W2)**
>
> > W2: In order to enhancing the key experts, the author of the paper proposes 5 strategy and the author of the paper uses strategy D. But the reader would be more convincing of why the author of the paper choose such design.
>
> Thank you for this helpful suggestion. In Section 4.3 of the paper, we showed **through experiments on MATH‑500, AIME2024, and AIME2025 that among the five proposed enhancement strategies, Strategy D achieves the best accuracy improvements, and therefore we used Strategy D** in subsequent experiments. We understand that the reader would like to see, beyond experiments, some **theoretical analysis** for this design choice, and we add the following (and include this new content in **Appendix A.10**, highlighted in blue in the revised version).
>
> In the following, we briefly summarize and refine the key points of our analysis.
>
> **(1) Initial motivation: Strategies A and B.**
> An initial intuition was straightforward — since key experts play a decisive role in task‑specific reasoning, one might expect model performance to improve if all tokens are explicitly routed through or enhanced by these key experts. Based on this idea, **Strategies A** and **B** were designed to enforce the participation of key experts for all tokens, strengthening their influence across the entire sequence. This represents the most direct attempt to leverage key experts’ strong impact on output generation.
>
> **(2) Why Strategies A and B are not stable.**
> Analyses in Sections 4.1 – 4.2 show that key experts are closely tied to domain‑specific tokens. For example, in the *math* domain, key experts are mainly activated by digits, letters, and operators, but not by unrelated words such as pronouns. Forcing all tokens to use key experts disregards this specialization and thus causes unstable or degraded performance. This behavior is **consistent with the experimental results in Section 4.3**, where **Strategies A** and **B** exhibited larger fluctuations and lower average accuracy. Therefore, a more adaptive mechanism is needed to selectively enhance key experts only when beneficial.
>
> **(3) Adaptive Enhancement (Strategies C and D) Yields Better Performance.**
> A natural idea might be to build a lookup table of task‑relevant tokens to explicitly control enhancement. However, such a method would be overly cumbersome and inconsistent with our goal of simplicity and efficiency. Instead, we empirically observed that for tokens within a key expert’s domain (e.g., math‑related tokens), the corresponding key experts often appear within the top‑2k routing weights even when they are not in the top‑k, while for unrelated tokens they almost never appear in that range. This provides a simple and effective criterion for deciding when and where to enhance domain‑specific experts.
>
> To verify this empirical observation, in **Appendix A.10** we further include a simple and effective experiment conducted on *Qwen3‑30B‑A3B* that examines the relationship between key experts and token types in the math domain. The results support our intuition that adaptive strategies can accurately associate key experts with domain‑relevant tokens, achieving selective enhancement without unnecessary activation **(The specific experimental setup and analysis details are provided in Appendix A.10).** And the experimental results in Section 4.3 demonstrate the effectiveness of Strategies C and D.
>
> **(4) Choice between Strategy C and D.**
> The main difference between **Strategies C and D** lies in whether the minimally weighted selected expert is replaced. As the least‑weighted expert contributes negligibly to model performance, replacing or retaining it makes little difference, **which is also consistent with the motivation of our Ban method**. For simplicity and implementation efficiency, we therefore adopted **Strategy D**.

---

> ### Author Response · Authors · 2025-11-17
> **Part 3/3 (W3)**
>
> > W3: Comparing to Pick, the Ban sometimes degrade the accuracy, which is understandable. But combine with Pick, the goal of this paper would become improve the performance of the MoE model. However, the Ban&Pick results in Table 3 shows varying improve. Although the difficulty of the tasks are different, some of the tasks like Math500 has limited improvement. And within the same task like GPQA, different models adapt to the Ban&Pick differently, which shows unstable performance of the method.
>
> This concern has two parts: instability across datasets and instability across models. We address them one by one below:
>
> 1. **Stability across datasets.**
>    First, we would like to clarify that **Ban** and **Pick** are two functionally independent modules. If one prioritizes accuracy improvements, Pick can be used on its own; as shown in Table 1, it brings a clear and sometimes substantial gain on MATH-500. When Ban is combined with Pick, on top of an already substantial inference speedup (exceeding 1.2×), we acknowledge that the additional accuracy gains can be limited on a few datasets. That said, we hope to emphasize that achieving both acceleration and accuracy gains simultaneously on fine-grained MoE is intrinsically challenging; to the best of our knowledge, existing MoE routing/pruning methods during inference typically trade one for the other rather than improving both. In this light, even if the gains are modest on some datasets, we believe the contribution remains practically meaningful.
>
> 2. **Stability across models.**
>    As Table 1 in the paper indicates, Pick tends to yield larger benefits on the **newer Qwen3-MoE** family than on the **earlier DeepSeek-V2** series. Our understanding is that Qwen3-MoE exhibits **stronger expert specialization**, which makes the identified key experts more influential. We also note that moving toward sparser routing with more pronounced expert specialization is a visible trend in fine-grained MoE design: newer models typically have higher sparsity and clearer specialization (e.g., Qwen3-Next with 512 experts per layer and Kimi-K2 with 384 experts per layer).
>
>    Qwen3-Next series were released in mid-September; shortly after our submission, we evaluated Pick on **Qwen3-Next-80B-A3B-Instruct** (max\_new\_tokens = 32k, temperature = 0.7, top-p = 0.8, top-k = 20), and the results (shown in the table below) aligned well with our hypothesis, showing an average **+3.8%** accuracy improvement across five datasets (higher than the **+2.83%** observed on the Qwen3-MoE series).
>
>    |     Tasks | AIME24 | MATH-500 | GPQA-Diamond | LiveCodeBench-V6 | HumanEval+ | Average |
>    | --------: | :----: | :------: | :----------: | :--------------: | :--------: | :-----: |
>    |  **Base** | 83.33  |  97.60   |    75.25     |      56.57       |   87.80    |  80.11  |
>    | **+Pick** | 90.00  |  98.80   |    79.29     |      60.00       |   91.46    |  83.91  |
>
>    **Based on this trajectory, we are cautiously optimistic that newer fine-grained MoE models will benefit even more from Pick. Therefore, rather than viewing these variations as instability, we regard the stronger synergy between our method and newer fine-grained MoE architectures as a positive indication for future applications.**

---

> ### Author Response · Authors · 2025-11-27
> **Looking forward to your feedback**
>
> Dear Reviewer Rchp,
>
> I hope this message finds you well. Thank you for your thoughtful review and the very meaningful suggestions you provided. Following your comments, in the revised version we have:
>
> 1. added a detailed analysis of the reasons behind the under‑utilization of key experts, and
> 2. provided theoretical analysis of the different enhancement strategies we designed in the Pick method.
>
> We believe these additions have helped improve the quality of the paper. In the rebuttal, we also explained and validated the observation that the Pick method performs even better on newer fine‑grained MoE models, which we hope adequately addresses your concerns regarding unstable performance.
>
> As the discussion period has entered its final week, we would like to ensure that we have addressed all your concerns satisfactorily. If you have any additional questions or feedback you would like us to consider, please feel free to let us know. Your feedback is extremely valuable to us, and we are eager to resolve any remaining issues to further strengthen our work.

---

### Official Review · Reviewer_2H1d · 2025-10-31

**Soundness:** 3
**Presentation:** 3
**Contribution:** 3
**Rating:** 4
**Confidence:** 3

**Summary:**

This paper claims that the routing strategies learned during pre-training cannot fully exploit this specialization, and thus proposes Pick and Ban strategies to optimizes expert utilization in inference. It improves both the accuracy and efficiency of fine-grained MoE without retraining or architectural changes.

**Strengths:**

- The proposed Ban&Pick is a post-training strategy. It does not require expensive retraining of the foundation model.
- The paper provides a thorough evaluation (Section 6) across multiple modern MoE models (DeepSeek-V2-Lite, DeepSeek-v2.5, Qwen3-30B, Qwen3-235B) and a diverse set of challenging benchmarks (AIME2024, GPQA, LiveCodeBench)

**Weaknesses:**

- Concern on generalization capability. The "Pick" module relies on identifying key experts for specific domains (math, code, general). This introduces a calibration step and raises questions about scalability. It's unclear how this would perform on tasks that are highly interdisciplinary or fall outside the tested domains.

- Hyperparameter Sensitivity. The method introduces new hyperparameters that may require tuning.

**Questions:**

How would the "Pick" module function on a task that intrinsically blends domains, such as writing a legal analysis (general reasoning) of a software patent (code)?

---

> ### Author Response · Authors · 2025-11-17
> **Part 1/3 (W1)**
>
> Dear reviewer 2H1d,
>
> Thank you for praising the lightweight design of our Ban&Pick framework, as well as the thorough evaluation of proposed methods. Below, we will address your questions one by one.
>
> > W1: Concern on generalization capability. The "Pick" module relies on identifying key experts for specific domains (math, code, general). This introduces a calibration step and raises questions about scalability. It's unclear how this would perform on tasks that are highly interdisciplinary or fall outside the tested domains.
>
> Thank you for raising this important question about Pick’s generalization. **This aspect is also a central consideration in our work.** We address it from two angles: feasibility in cross-domain settings and coverage for tasks beyond those explicitly discussed in the paper.
>
> 1. **Cross-domain feasibility.**
>    Regarding cross-domain feasibility, Section 7 of the paper provides a detailed empirical study of Pick in multi-domain settings. When we jointly enhance key experts from multiple domains, the gain on any single domain may become slightly smaller than when enhancing that domain alone; **however, the overall accuracy across domains is still significantly improved**, demonstrating that Pick remains effective in interdisciplinary scenarios. **Additionally, we would also like to clarify that the “general reasoning” domain referenced in the paper is in fact a broad class of tasks.** As illustrated by the word cloud in Figure 2, general-domain key experts are associated with branch tokens that are useful for generic reasoning, which can benefit a wide range of tasks. Consistent with this, Table 5 in Section 7 shows that enhancing general-domain key experts alone not only improves GPQA-Diamond by 3.53%, but also yields gains on AIME24 (+1.33%, math) and HumanEval+ (+1.22%, code). Taken together, these results provide evidence that Pick can operate effectively in cross-domain contexts.
>
> 2. **Beyond the tested domains.**
>    For tasks not explicitly covered in the main experiments, given the breadth of “general” noted above, we conducted an extra evaluation where we enhanced only key experts for the “general” domain on Qwen3-30B-A3B, and then tested this setting on two additional benchmarks that were not included in the original experiments: C-Eval (a cross-disciplinary benchmark spanning 52 categories, including economics and history) and IFEval (an alignment task) to further assess the generalization of Pick beyond the originally tested domains.
>
>    |  Task  | Setting | Accuracy (%) |
>    | :----: | :-----: | :----------: |
>    | C–Eval |  Base   |    86.60     |
>    | C–Eval | + Pick  |    88.34     |
>    | IFEval |  Base   |    85.95     |
>    | IFEval | + Pick  |    87.80     |
>
>    The results indicate that enhancing general-domain key experts also improves performance on these broader task families. In light of these findings—and in combination with the math and code domains—we believe Pick already covers a substantial portion of real-world workloads. **That said, if there exist niche domains not examined in our study where Pick brings limited benefit, we hope this can be viewed as an acceptable trade-off for a lightweight, plug-and-play approach that requires no retraining and minimal integration effort.**

---

> ### Author Response · Authors · 2025-11-17
> **Part 2/3 (W2)**
>
> > W2: Hyperparameter Sensitivity. The method introduces new hyperparameters that may require tuning.
>
> Thank you for pointing out the issue of hyperparameter sensitivity. We agree that this is important for practical usability. Although our method introduces two new hyperparameters—Kmin and λ in Ban, **we would like to clarify that these hyperparameters do not require expensive tuning and empirically exhibit strong robustness.** The corresponding evidence and analyses are provided in Appendix A.5 of the original paper; we briefly summarize them below.
>
> 1. **Kmin.**
>    As discussed in the paper, Kmin guarantees a minimum number of selected experts. **This is not intended as a finely tuned optimal value, but rather as a fixed configuration specific to a given fine‑grained MoE model.** For example, Table 7 in A.5 shows that Qwen3‑30B‑A3B largely maintains performance when the number of selected experts is 4 or 3, but collapses at 2; hence we set Kmin = 3. In practice, identifying such a threshold does not require extensive reasoning benchmarks: quick zero‑shot multiple‑choice sets such as PIQA or BoolQ are sufficient to reach the same conclusion. Therefore, no costly tuning is needed.
>
> 2. **λ.**
>    In Ban, λ controls the sparsity level of expert pruning. **We note that virtually all sparsification methods for acceleration involve a sparsity‑control hyperparameter; the key consideration is robustness.** Table 8 in A.5 demonstrates that λ is robust on Qwen3‑30B‑A3B: within λ ∈ [0.7, 0.9], both accuracy and speedup remain stable with only small fluctuations. Even a conservative choice of λ = 0.9 achieves about 1.2× speedup in vLLM while preserving accuracy, suggesting that practical deployment does not require heavy tuning.
>
>    To further verify the robustness of λ across different model scales, we additionally conducted experiments on **Qwen3‑235B‑A22B**. We varied λ in {0.7, 0.8, 0.9} and evaluated on the same setting like in A.5. The results (see table below) demonstrate that both accuracy and speedup remain stable across settings, further suggesting that tthe method is robust to λ.
>
>    | \(\lambda\) | Avg Top‑k | AIME2024 | GPQA  | HumanEval+ | Speedup |
>    | ----------- | --------- | :------: | :---: | :--------: | :-----: |
>    | (no Ban)    | 8.00      |  84.67   | 71.21 |   78.05    |  1.00   |
>    | 0.7         | 4.77      |  83.33   | 69.70 |   75.61    |  1.27   |
>    | 0.8         | 5.03      |  83.33   | 70.71 |   76.22    |  1.25   |
>    | 0.9         | 5.21      |  83.33   | 70.20 |   76.83    |  1.24   |
>
>
> **Taken together, while our method does introduce hyperparameters, they are straightforward to set, empirically robust, and do not necessitate substantial tuning effort in practice.**

---

> ### Author Response · Authors · 2025-11-17
> **Part 3/3 (Q1)**
>
> > Q1: How would the "Pick" module function on a task that intrinsically blends domains, such as writing a legal analysis (general reasoning) of a software patent (code)?
>
> Thank you for this question, which is closely related to the practical usability of **Pick**. **We want to clarify that in Pick, enhancing key experts of one domain rarely harms performance in other domains, and jointly enhancing key experts from multiple domains can yield simultaneous improvements across them. And In Section 7 of the paper, we validated this property through concrete multi-domain experiments.** The results show that when key experts from multiple domains are enhanced simultaneously, **the model achieves substantial performance gains across all domains**, though the improvement within each individual domain is slightly smaller than when enhancing that domain alone. For example, when we jointly enhance key experts for the general and code domains, we still obtain improvements of **+2.52%** on *GPQA‑Diamond* (general) and **+3.66%** on *HumanEval+* (code). We believe these results support the expectation that **Pick** can also improve performance on tasks that intrinsically blend domains, such as writing a legal analysis (general reasoning) of a software patent (code).
>
> **In practical use, when the target task spans multiple domains, one can simply enhance the corresponding key experts of those domains simultaneously, as the mutual influence among these enhancements is very minor. Moreover, in scenarios where the task type cannot be known in advance, one can directly choose to enhance the key experts of the math, general, and code domains simultaneously; as shown in Table 5 of the paper, this configuration still yields accuracy improvements of 2.02% on GPQA‑Diamond (general), 3.05% on HumanEval+ (code), and 3.33% on AIME2024 (math), demonstrating the practical usability and robustness of the Pick method.**

---

> ### Author Response · Authors · 2025-11-27
> **Looking forward to your feedback**
>
> Dear Reviewer 2H1d,
>
> I hope this message finds you well. Thank you for your thoughtful review and the very meaningful concerns you raised. In response to your comments regarding the generalization capability of our Ban&Pick framework and its hyperparameter sensitivity, we have carefully prepared detailed explanations and analyses to address these points.
>
> We hope that these additions clarify the behavior of our method and resolve the issues you highlighted.
>
> As the discussion period is nearing its end, we would like to ensure that all your questions have been fully addressed. If there are any remaining issues or further feedback you would like us to consider, please feel free to let us know. Your feedback is highly valuable to us, and we are committed to improving our work in every possible way.

---

### Author Response · Authors · 2025-12-01
**Summary of Rebuttal**

**Dear Area Chair and Reviewers,**

We would first like to sincerely thank the **Area Chair** for the tremendous effort. To help ease your burden, we have carefully and objectively prepared the following **summary** to assist you in quickly understanding how each reviewer evaluated our paper and how we addressed every concern and suggestion during the rebuttal stage to improve the paper’s quality.

---

### **Reviewer 2H1d**

Reviewer 2H1d positively acknowledged the **lightweight “Ban&Pick” framework** that requires no additional training, along with the **comprehensive experiments** verifying its effectiveness. Their main concerns focused on (1) the **generalization ability** of the Pick module—especially its behavior in multi-domain tasks—and (2) the need to **tune the hyperparameters** introduced in Ban.

For the first concern, we clarified that **Section 7 in the original paper already provides experimental evidence** showing Pick’s effectiveness under multi-domain settings. In the rebuttal, we referenced these results and **added two new experiments** on datasets covering wider domains beyond those originally tested, further supporting Pick’s generalization capability.

Regarding the hyperparameter issue, we explained that although Ban introduces two hyperparameters (*K*ₘᵢₙ and λ), they are **robust and easy to set**, thus requiring no costly tuning. We supported this claim with the analysis and experiments in **Appendix A.5 of the original paper**, and we further included **additional robustness experiments** across models to reinforce our point.

---

### **Reviewer Rchp**

Reviewer Rchp praised our **detailed analysis of key experts**, **clear writing**, and the **simplicity and effectiveness** of the Ban&Pick framework, describing the findings as *novel and inspiring*. We are very grateful for these encouraging comments and for the two constructive suggestions proposed in the weaknesses section:
(1) to provide a **theoretical discussion explaining why key experts are under-utilized**, and
(2) to include **analysis on the rationale behind the design choices** in enhancement strategies.
We fully agreed with both suggestions. Accordingly, in response to the first suggestion, we added a detailed theoretical analysis in **Section 8** of the revised version, explaining how premature routing and the auxiliary balancing loss jointly lead to insufficient emphasis on key experts. For the second suggestion, we incorporated an in‑depth discussion in **Appendix A.10**, analyzing the design principles and reasoning behind the different enhancement strategies. **These additions have significantly strengthened the theoretical grounding and interpretability of our work.** We regret that, due to the early closure of discussion, we could not further exchange ideas with the reviewer.

Additionally, Reviewer Rchp raised a concern regarding the stability of performance across different models. To address this, we clarified that newer fine-grained MoE architectures exhibit stronger expert specialization, making our method more effective on them. We supported this observation by adding experiments on the **latest Qwen3‑Next series**, which confirmed greater improvements.

---

### **Reviewer 364k**

Reviewer 364k commended the **thorough analysis and validation**, the **lightweight design** of Ban&Pick, and their **significant performance gains** and **practical applicability** in real-world scenarios. The reviewer suggested (1) providing a theoretical explanation for the mechanism of key experts, and (2) including a **system-level performance analysis**.

Since the first suggestion closely aligns with Rchp’s comments, we integrated both into the newly added theoretical analysis in **Section 8** of the revised version. For the second suggestion, we conducted and included new experiments measuring throughput and memory usage, which have been added to **Appendix A.11** of the revised version, further enriching the paper’s evaluation.

Reviewer 364k carefully pointed out several typos and suggested the inclusion of pseudo-code. We have corrected these typos in the revised version and provided the corresponding content in Appendix A.9. Furthermore, we actively discussed the open-ended questions raised by the reviewer, such as the potential integration of Ban & Pick with other techniques and the possibility of task-specific enhancements.

---

### **Summary**

Although the discussion was unfortunately closed before receiving all reviewers' feedback, we deeply appreciate the reviewers’ **time, efforts, and valuable suggestions**. We believe our rebuttal has **earnestly addressed all the raised concerns**, and through their guidance, we have **made substantial improvements** in both the technical depth and presentation quality of our paper.

Finally, we once again express our sincere gratitude to the **Area Chair** for the exceptional work and responsibility taken on under these unusual circumstances.

The Authors

---

### Meta-Review · Area_Chair_m7Fd · 2026-01-07

**Summary:**

The paper proposes a lightweight, post-training routing strategy for fine-grained MoE models and presents extensive empirical evaluations across several benchmarks. Reviewers generally acknowledge the practical motivation, plug-and-play design, and potential efficiency gains, as well as the breadth of experiments.

However, multiple core concerns raised in the initial reviews remain insufficiently resolved under a conservative assessment. In particular, questions persist regarding the generalization of the proposed method beyond the evaluated domains, the isolation of the method’s contribution from model- or architecture-specific effects, and the stability and consistency of gains across tasks and model families. Requests for stronger theoretical grounding, clearer justification of key design choices, and broader or more appropriate comparative baselines were not explicitly acknowledged as resolved by all reviewers. Reproducibility-related clarity (e.g., sensitivity to design and hyperparameter choices) also remains a concern.

Given the interruption of the review process and limited post-rebuttal participation, this assessment is based strictly on the available materials, without assuming any post-rebuttal score changes or implicit resolution of outstanding issues.

**Reviewer Concerns:**

Reviewer 2H1d.
The reviewer raised concerns about the generalization of the Pick module to interdisciplinary or out-of-domain tasks, as well as potential hyperparameter sensitivity. The rebuttal provides additional empirical evidence and clarifications addressing these points, including multi-domain evaluations and robustness analyses. However, since the reviewer did not participate in the post-rebuttal discussion, these concerns are considered partially addressed but not conclusively resolved.

Reviewer Rchp.
The reviewer requested a clearer causal explanation for why key experts are under-utilized, stronger justification for the design choice among enhancement strategies, and clarification on performance stability across models and datasets. The rebuttal adds theoretical discussion, design rationale, and additional experimental results. In the absence of post-rebuttal feedback from the reviewer, these concerns are assessed as partially addressed, with some residual uncertainty remaining.

Reviewer 364k.
The reviewer suggested deeper theoretical analysis, broader system-level evaluation, and comparisons with recent MoE optimization baselines, along with minor presentation issues. The rebuttal adds theory, system-level metrics, and addresses presentation concerns. However, the lack of direct comparisons to newer baselines remains an outstanding concern, as explicitly noted by the reviewer, who indicated that their evaluation remained unchanged.

**Reviewer Scores:**

Reviewer 2H1d.
Initial score: 4. Given the absence of explicit post-rebuttal feedback from the reviewer, and following a conservative policy, the score is assumed to remain unchanged.

Reviewer Rchp.
Initial score: 4. Although the rebuttal substantively engages with the reviewer’s concerns, the reviewer did not participate further in discussion. Conservatively, the score is assumed to remain unchanged.

Reviewer 364k.
Initial score: 8. The reviewer explicitly stated that they had read the rebuttal and that their review and score remain unchanged.

---

### Decision · Program_Chairs · 2026-01-26

Reject